# Diffusion on Graph: Augmentation of Graph Structure for Node Classification

**Yancheng Wang**                                                   *yancheng.wang@asu.edu*
**Changyu Liu**                                                        *changyu2@asu.edu*
**Yingzhen Yang**                                                   *yingzhen.yang@asu.edu*
*School of Computing and Augmented Intelligence*
*Arizona State University*

**Reviewed on OpenReview:** *https://openreview.net/forum?id=tzW948kU6x*

## Abstract

Graph diffusion models have recently been proposed to synthesize entire graphs, such as molecule graphs. Although existing methods have shown great performance in generating entire graphs for graph-level learning tasks, no graph diffusion models have been developed to generate synthetic graph structures, that is, synthetic nodes and associated edges within a given graph, for node-level learning tasks. Inspired by the research in the computer vision literature using synthetic data for enhanced performance, we propose Diffusion on Graph (DoG), which generates synthetic graph structures to boost the performance of GNNs. The synthetic graph structures generated by DoG are combined with the original graph to form an augmented graph for the training of node-level learning tasks, such as node classification and graph contrastive learning (GCL). To improve the efficiency of the generation process, a Bi-Level Neighbor Map Decoder (BLND) is introduced in DoG. To mitigate the adverse effect of the noise introduced by the synthetic graph structures, a low-rank regularization method is proposed for the training of graph neural networks (GNNs) on the augmented graphs. Extensive experiments on various graph datasets for semi-supervised node classification and graph contrastive learning have been conducted to demonstrate the effectiveness of DoG with low-rank regularization. The code of DoG is available at https://github.com/Statistical-Deep-Learning/DoG.

## 1 Introduction

Diffusion models (Song et al., 2021c; Song & Ermon, 2019a; Ho et al., 2020) have gained prominence as a class of generative models capable of producing high-quality synthetic data (Gao et al., 2023; Rombach et al., 2022; Baranchuk et al., 2022). Building upon the success of diffusion models, conditional diffusion models (Rombach et al., 2022; Ho & Salimans, 2022) utilize external information such as text descriptions, class labels, or images in the noising and denoising phases, steering the generation process towards contextually relevant outcomes. Building on these advancements, recent studies (Trabucco et al., 2023; Azizi et al., 2023) employ class-conditional diffusion models to produce labeled synthetic data, which is then incorporated into the training datasets of classification tasks as a form of data augmentation. Following the success of diffusion models in the visual and textual fields, graph diffusion models (Niu et al., 2020; Jo et al., 2022; Haefeli et al., 2022; Vignac et al., 2023a; Limnios et al., 2023) have recently been studied to tailor diffusion models to tackle the specific challenges of synthesizing graph-structured data. This adaptation has broadened the scope of synthetic graph generation, allowing for the creation of complex, realistic graphs that preserve the fundamental characteristics of real-world datasets. Although current models are predominantly aimed at generating complete graphs (Niu et al., 2020; Jo et al., 2022; Haefeli et al., 2022) or small graphs (Vignac et al., 2023a; Limnios et al., 2023) resembling the graphs in the given graph datasets for graph-level learning tasks, there remains an unexplored potential in generating synthetic graph structures

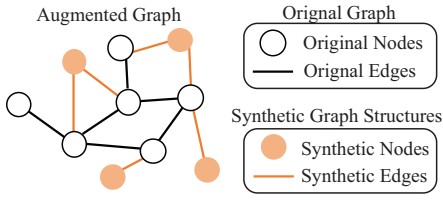

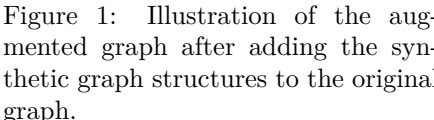

Figure 1: Illustration of the augmented graph after adding the synthetic graph structures to the original graph.

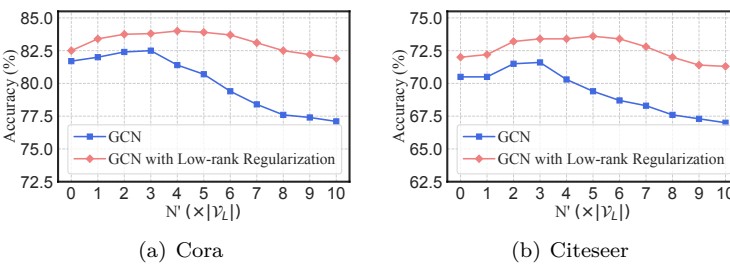

(a) Cora        (b) Citeseer

Figure 2: Node classification accuracy of GCN on Cora and Citeseer trained with different numbers ($N'$) of synthetic nodes added. $\mathcal{V}_L$ is the set of labeled training nodes in the original graph.

for node-level classification tasks within a single graph. The synthetic graph structure refers to both the synthetic nodes and the edges connecting them within the same graph as illustrated in Figure 1.

In this work, we focus on node-level generation, aiming to synthesize graph structures within a given graph. Limited training data is a common issue in graph datasets. For example, only 140 of 3327 nodes in Citeseer (Sen et al., 2008b) are in the training set. Inspired by the synthetic data augmentation methods from the computer vision literature (Azizi et al., 2023; He et al., 2022), we propose a new research direction of using synthetic graph structures to increase the number of training nodes and thereby improve the node classification performance. Figure 2 illustrates that the performance of GCN trained on augmented graphs increases with suitably more synthetic structures added to the training set.

We propose a novel Diffusion Model on Graph (DoG), which, to the best of our knowledge, is the first graph diffusion model to generate synthetic graph structures comprising both synthetic nodes and the synthetic edges associated with the synthetic nodes within a single given graph. DoG comprises a novel Graph Autoencoder (GAE) and a Latent Diffusion Model (LDM). The encoder of the GAE encodes node attributes and edges connecting to the nodes into a continuous latent space, and the decoder of the GAE decodes latent features to node attributes and edges connecting to the nodes. The LDM aims at synthesizing the latent features of the synthetic graph structures. When trained with Classifier-free Guidance (CFG) (Ho & Salimans, 2022), the LDM can generate the latent features of the synthetic graph structures where the node belongs to a specific class. The motivation for leveraging the LDM in processing the latent representations primarily stems from its ability to enhance the generative capabilities of the diffusion model while ensuring computational efficiency. The LDM trains the diffusion model in the compressed latent features generated by the GAE rather than directly in the high-dimensional input data space. This approach significantly reduces the dimensionality of the data being processed, which decreases the computational resources required for the training and the inference of the diffusion model. Moreover, the LDM has shown remarkable performance in generating high-quality synthetic data across various domains (Rombach et al., 2022; Lovelace et al., 2023). Moreover, decoding the neighbors of a node from the latent space in the entire graph can be computationally expensive, considering that a node can potentially connect to any other nodes within the graph as its neighbors. To overcome this issue, our DoG features a Bi-Level Neighborhood Decoder (BLND), which reconstructs the edges connecting to a node in a hierarchical manner. The nodes in the graph are first divided into several clusters using a balanced clustering method. For each synthetic node, instead of finding the neighbors among all the nodes in the original graph, BLND first finds the clusters containing the original nodes to which the synthetic node should connect through an inter-cluster decoder. Then, BLND identifies the individual nodes within each of the identified clusters by generating an intra-cluster neighbor map for that specific cluster with an intra-cluster decoder. Combining the synthetic graph structures generated by DoG with the original graph, we obtain an augmented graph, illustrated in Figure 1, for the downstream node-level learning tasks.

Due to the inherent randomness in the forward process of diffusion models (Ho et al., 2020) and the difficulty in accurately conditioning the LDM with class signals (Fu et al., 2024), the synthetic graph structures inevitably contain noise. Consequently, training node classifiers directly on these augmented graphs may result in degraded performance (Azizi et al., 2023). Let $\mathcal{V}_L$ denote the set of labeled training nodes in the

original graph. Let $N'$ denote the number of synthetic nodes generated by our DoG, which measures the amount of synthetic graph structures added to the augmented graph. As illustrated in Figure 2, while the performance of GCN trained on augmented graph increases with more synthetic graph structures added when $N' < 3|\mathcal{V}_L|$, more synthetic graph structures added to the augmented graph hurts the performance of the vanilla GCN severely when $N' > 3|\mathcal{V}_L|$. As a result, it remains an important and interesting problem to effectively train GNNs on the augmented graph, which can be very noisy due to the noise in synthetic graph structures. To this end, we propose a low-rank learning method inspired by the low frequency property illustrated in Figure 5 in Section C of the appendix. The low frequency property suggests that the low-rank part of features learned by a GNN cover the majority of the information of the training labels. Motivated by this observation, we add a low-rank regularization term in the training loss of a GNN for node-level learning tasks, such as node classification and graph contrastive learning, on the augmented graph. The low-rank regularization term encourages that only the low-rank part of the features learned by a GNN is used for classification, so that the noise in the synthetic graph structures in the high-rank part of the features would mostly not affect the performance of the GNN. We evaluate the semi-supervised node classification accuracy of GCN trained with low-rank regularization and regular GCN with different amounts of synthetic graph structures added to the augmented graph. As illustrated in Figure 2, the GCN trained with the low-rank regularization constantly outperforms the vanilla GCN and exhibits more stable performance than the vanilla GCN with different amount of synthetic nodes added to the augmented graph. Such results suggest that our low-rank learning method significantly reduces the adverse effect of the noise in the augmented graph when more synthetic structures are introduced.

## 1.1 Contributions

First, we propose a diffusion model on graph-structured data, termed Diffusion on Graph (DoG). To the best of our knowledge, DoG is the first node-level graph diffusion model that synthesizes nodes and associated edges in a single given graph. In contrast, previous graph diffusion models focused on generating entire graphs by learning diffusion models from graph-level learning datasets. Our DoG model also features a novel Bi-Level Neighborhood Decoder (BLND), which efficiently reconstructs the edges connecting to a node in a bi-level hierarchical manner.

Second, we propose a low-rank learning method that trains GNNs or GCL models on the augmented graph with low-rank regularization, which largely mitigates the adverse effect of the noise from the synthetic graph structures. The low-rank learning method uses a low-rank regularization term in the usual training loss by cross-entropy, which is inspired by the low frequency property illustrated in Figure 5 in Section C of the appendix. Our low-rank learning method enjoys both strong theoretical guarantee in terms of the error bound for the test loss presented in Section A of the appendix, and compelling empirical results. Extensive experiment results in Table 1 and Table 2 of training GNNs and GCL models on augmented graphs with low-rank regularization suggest that the low-rank learning method effectively reduces the adverse effect of noise in the synthetic graph structures. In addition, results in Table 3 show that the low-rank regularization method significantly outperforms existing data selection methods for learning with noisy data.

Different from existing node-level graph augmentation methods reviewed in Section 2.2, DoG is among the first to generate synthetic graph structures which improve the prediction accuracy of GNNs for node classification by enlarging the size of the training set. The augmentation method of DoG is orthogonal to existing node-level data augmentation methods. For example, existing node-level augmentation methods, such as GraphMix (Verma et al., 2021), FLAG (Kong et al., 2022), and NODEDUP (Guo et al., 2024), can be directly applied to the training of GNNs on the graphs augmented by DoG with enhanced performance. Results in Table 4 show that DoG can significantly improve the performance of models trained with GraphMix, FLAG, and NODEDUP. As a result, DoG is expected to generate broad impact in the graph learning literature as a new graph augmentation method.

## 2 Related Works

### 2.1 Diffusion Models

Score-based diffusion models, which smoothly transform data to noise using a diffusion process and synthesize samples by learning and simulating the time reversal of this diffusion (Song et al., 2021c), have demonstrated

superior performance across a wide range of generation tasks (Ho et al., 2020; Song & Ermon, 2019a; Gao et al., 2023; Rombach et al., 2022; Baranchuk et al., 2022). As DDPM suffers from large computational overhead in the sampling process, many recent works (Song et al., 2021d;b; Song & Ermon, 2020) focus on accelerating DDPM. For instance, LDM (Rombach et al., 2022) significantly improves the computational efficiency and the quality of generated images by training DDPM on the latent features of the training data encoded by pre-trained autoencoders. Recently, graph diffusion models have also been widely studied to generate synthetic graphs (Niu et al., 2020; Jo et al., 2022; Haefeli et al., 2022; Jo et al., 2022; Zhou et al., 2024). Early works (Jo et al., 2022; Haefeli et al., 2022; Vignac et al., 2023a) on graph diffusion seek to design discrete diffusion models tailored to the discrete nature of the adjacency matrix of graph data. Inspired by LDM (Rombach et al., 2022), SaGess (Limnios et al., 2023) embeds the graph structures and features of entire graphs into latent features and trains a DDPM on the latent feature space. Albeit the above efforts have been made to adopt diffusion models to generate synthetic graphs, all existing graph diffusion models are designed for graph-level synthetic data generation, which cannot be generalized to node-level generation within a single graph. In contrast, our work aims to generate synthetic graph structures, that are, synthetic nodes and the associated edges at the same time for node-level learning tasks.

## 2.2 Generative Data Augmentation and Data Selection

**Node-level Graph Data Augmentation.** Node-level graph data augmentation methods enhance the training of GNNs on graphs by modifying the structure, features, or labels of nodes in the graph (Zhao et al., 2023). Structure augmentation methods (Gasteiger et al., 2019; Zhao et al., 2021; Rong et al., 2020; Feng et al., 2022) modify the connectivity of nodes in the graph by either adding or removing edges. TADA (Lai et al., 2024) employs the attribute-aware graph structure sparsification to augment the graph structure for the training of the GNNs. Feature augmentation methods (You et al., 2020; Kong et al., 2022) randomly alter the node features during the training process. For instance, FLAG (Kong et al., 2022) iteratively augments node features with gradient-based adversarial perturbations during training to improve the performance of GNNs at the test time. LGGD (Azad & Fang, 2024) proposes to generate new features for the training nodes in the given graph based on the generalized geodesic distance as data augmentation for the node classification task. Label augmentation methods adjust the labels of nodes in the training set. For example, Mixup-based methods (Han et al., 2022; Wang et al., 2021; Verma et al., 2021) such as GraphMix (Verma et al., 2021) create interpolations of existing labeled nodes for the training of GNNs. GeoMix (Zhao et al., 2024) introduces a novel Mixup approach based on the in-place graph editing to augment the training set in the given graph for the node classification task. NODEDUP (Guo et al., 2024) is the first work that enlarges the training set of the graph by duplicating the nodes in it.

We remark that the node-level data augmentation methods, which modify the structure, features, or labels of nodes in the graph surveyed above, are orthogonal to DoG. As evidenced in Table 4 in Section 5.3, combining the DoG with the existing node-level data augmentation methods further boosts the performance of the existing node-level data augmentation methods and the DoG.

**Generative Data Augmentation.** Synthesizing informative training data as data augmentation for improving the performance of deep neural networks remains a vital yet challenging research area. Existing works predominantly focus on synthesizing training data through pre-trained deep generative models, such as Generative Adversarial Networks (GANs) (Zhang et al., 2021; Li et al., 2022) and diffusion models (He et al., 2022; Tian et al., 2023; Yuan et al., 2022; Bansal & Grover, 2023; Vendrow et al., 2023). For instance, recent works(Sarıyıldız et al., 2023; Lei et al., 2023; Azizi et al., 2023) leverage conditional latent diffusion models, such as Stable Diffusion(Saharia et al., 2022) to create class-conditioned samples for training image classification models.

**Training on Noisy Synthetic Data.** Despite the promising results of introducing synthetic data into the training of classification models, it is well-known that synthetic data generated by diffusion contain noise (Azizi et al., 2023; Trabucco et al., 2023; Na et al., 2024), which attributed to the randomness of the forward process (Ho et al., 2020) and the difficulty in accurately conditioning the LDM with class signals (Fu et al., 2024). To mitigate this issue, recent works (Sarıyıldız et al., 2023; Lei et al., 2023; Zhou et al., 2023) data selection methods, TopoFilter (Wu et al., 2020) and NGC (Wu et al., 2021) employ prompt engineering to generate synthetic data with improved quality for training downstream tasks(Sarıyıldız et al., 2023; Lei

et al., 2023; Zhou et al., 2023). However, these methods are sensitive to the design of prompts and are limited to multi-modal generative models. Beyond directly improving the quality of generated synthetic data, methods such as data selection (Han et al., 2018; Yu et al., 2019; Song et al., 2021a; Nguyen et al., 2020; Song et al., 2023), which selects a high-quality subset from the noisy training data to improve the performance of deep learning models, can also be used to select high-quality synthetic data for the training of downstream tasks. To this end, we also survey data selection methods for learning with noisy data. To demonstrate the superiority of our method in learning with noisy synthetic graph structures, we adapt data selection methods, TopoFilter (Wu et al., 2020) and NGC (Wu et al., 2021) to the settings of graph learning and establish baselines that use them to select a high-quality subset of the synthetic nodes. We then train GNNs on the original graph combined with the synthetic graph structures selected by these data selection methods.

## 2.3 Low-Rank Learning

Low-rank learning methods (Candès et al., 2011) have emerged as pivotal tools to mitigate noise, and enhance representation learning. For instance, (Yang & Cohen, 2015) introduced the technique of singular value pruning, which applies low-rank constraints within neural network layers to improve both performance and computational efficiency. Following that, (Hu et al., 2013) adopts Truncated Nuclear Norm Regularization (TNNR) to perform selective minimization of singular values to precisely recover the low-rank matrices under noisy conditions. The integration of TNNR in tensor completion tasks has also seen notable improvements, particularly through the application of tensor singular value decomposition (t-SVD) (Liu et al., 2017; Zhang et al., 2020). Following that, a recent study (Indyk et al., 2019) further improved the low-rank approximation methods for better practical efficacy. In the graph domain, low-rank learning methods have also been widely studied to improve the robustness of GNNs against noise and adversarial attacks (Savas & Dhillon, 2011; Entezari et al., 2020; Jin et al., 2020; Alchihabi et al., 2023; Xu et al., 2021).

## 2.4 Graph Neural Networks

Graph Neural Networks (GNNs) emerged as a powerful tool for learning expressive node representations for graph data. Most prevailing GNNs are built to learn the representations of nodes in a neighborhood aggregation manner, usually referred to as Message Passing Neural Networks (MPNNs). The representation of a node is learned by transforming and aggregating node features from its neighbors. Following this scheme, different GNN architectures have been proposed, including graph convolutional networks (GCNs) (Kipf & Welling, 2017), attention-based GNNs (Veličković et al., 2018; Wang et al., 2019; Iyer et al., 2021), and the others (Wu et al., 2019; Hamilton et al., 2017; Xu et al., 2019; Wu et al., 2019; Chen et al., 2020). Following the success of Transformers in natural language processing (Vaswani et al., 2017), transformer architectures have also been adapted to design GNNs to capture node-wise correlations (Wu et al., 2022; 2023b;a). Recently, GNNs, such as Graph Variational Autoencoders (GVAEs), have been adopted for graph-level synthetic graph generation, aiming to generate a set of synthetic graphs resembling the graphs in the given graph datasets (Mitton et al., 2020; Bongini et al., 2021; Zhu et al., 2022; Vignac et al., 2023b; Cai et al., 2024). Since graph-level synthetic graph generation methods cannot generate synthetic nodes within a given graph, they cannot be applied to generate synthetic labeled nodes to enlarge the training set of a graph for node classification. In contrast, our work focuses on node-level synthetic graph structure generation, which aims to generate synthetic nodes and synthetic edges connecting the synthetic nodes to the original nodes in a given graph.

## 3 Preliminaries

We first establish the notations for the attributed graph in Section 3.1. Furthermore, a comprehensive overview of Diffusion Models (DMs), Latent Diffusion Models (LDMs), and Classifier-free Guidance (CFG) is provided in Section B of the appendix.

## 3.1 Attributed Graph and Notations

An attributed graph with $N$ nodes is denoted by $\mathcal{G} = (\mathcal{V}, \mathcal{E}, \mathbf{X})$. Here, $\mathcal{V} = \{v_1, v_2, \ldots, v_N\}$ represents the nodes, and $\mathcal{E} \subseteq \mathcal{V} \times \mathcal{V}$ represents the edges. Node attributes are given by $\mathbf{X} \in \mathbb{R}^{N \times D}$, where each row

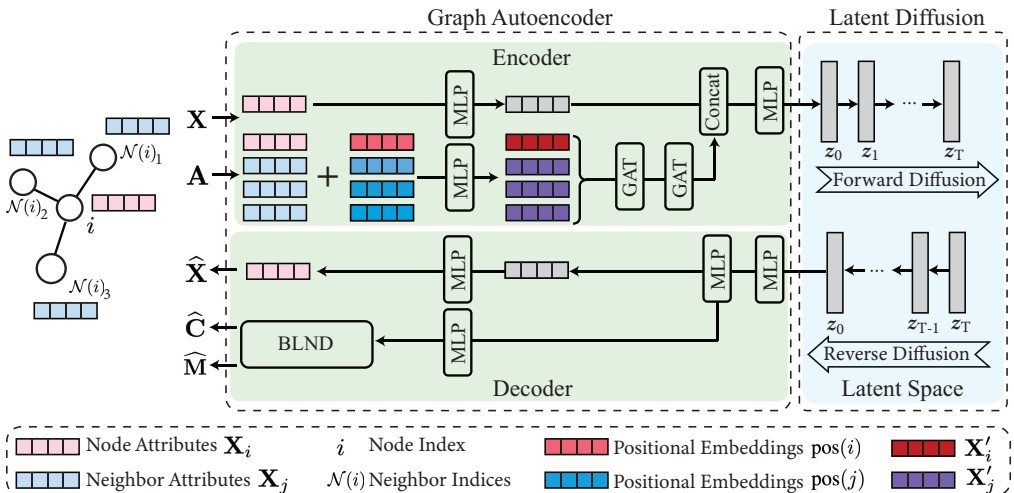

Figure 3: The overall framework of synthetic node generation process. The structure of the Bi-level Neighbor Map Decoder (BLND) is illustrated in Figure 4.

$\mathbf{X}_i \in \mathbb{R}^D$ corresponds to the attributes of node $v_i$. The adjacency matrix $\mathbf{A} \in \{0,1\}^{N \times N}$ of graph $\mathcal{G}$ defines connections such that $\mathbf{A}_{ij} = 1$ if $(v_i, v_j) \in \mathcal{E}$. Each row $\mathbf{A}_i$ represents the connections of node $v_i$. An augmented adjacency matrix, $\tilde{\mathbf{A}} = \mathbf{A} + \mathbf{I}$, includes self-loops. The diagonal degree matrix of this augmented matrix is $\tilde{\mathbf{D}}$. The neighborhood $\mathcal{N}(i) = \{j \mid \tilde{\mathbf{A}}_{i,j} = 1\}$ includes node $v_i$ itself and all nodes connected to $v_i$. The notation $[N]$ denotes all natural numbers from 1 to $N$ inclusive. $[\mathbf{A}]_i$ stands for the $i$-th row of a matrix $\mathbf{A}$. $\|\cdot\|_p$ denotes the $p$-norm of a vector or a matrix.

## 4 Methods

### 4.1 Diffusion on Graphs (DoG)

**Graph Autoencoder (GAE).** To encode a node $v_i$ into a latent feature with the encoder of the Graph Autoencoder (GAE), we first generate a latent feature of the node attribute $\mathbf{X}_i$ as $f(\mathbf{X}_i)$, where $f(\cdot)$ is a Multi-Layer Perceptron (MLP) layer, which consists of a linear layer followed by an activation function ReLU. To incorporate the information from the edges connected to $v_i$, we add positional embeddings to the node attributes of $v_i$'s neighbors. Positional embedding methods (Ma et al., 2021; You et al., 2019) have been used in GNNs to encode the global positional information of the nodes in a graph. In our work, the positional indexes are used to encode the positions of neighboring nodes of nodes in the graph into the latent space. For each neighbor $j \in \mathcal{N}(i)$, we modify the node attributes as $\mathbf{X}'_j = \mathbf{X}_j + \text{pos}(j)$, where $\text{pos}(\cdot)$ is a function converting the position index into an embedding vector (Vaswani et al., 2017). We apply two Graph Attention Network (GAT) (Veličković et al., 2018) layers to aggregate the information in $\{\mathbf{X}'_j \mid j \in \mathcal{N}(i)\}$ into a single latent feature $\mathbf{Z}'_i$. Next, we concatenate $\mathbf{Z}'_i$ with $f(\mathbf{X}_i)$ to obtain the latent feature of node $v_i$ by $\mathbf{Z}_i = f'(\mathbf{Z}'_i \| f(X))$, where $f'$ is another MLP layer encoding the concatenated features to the latent space of LDM with lower dimension $D'$. After encoding a node $v_i$ in the graph to latent feature $\mathbf{Z}_i$, the decoder of the GAE reconstructs the node attribute $\widehat{\mathbf{X}}_i$ and its associated edges $\widehat{\mathbf{A}}_i$. The node attributes $\widehat{\mathbf{X}}_i$ are reconstructed by three consecutive MLP layers. The edges connected to the node are reconstructed by the Bi-Level Neighborhood Decoder (BLND) introduced below. The overall framework of DoG is illustrated in Figure 3.

**Bi-Level Neighborhood Decoder (BLND).** Decoding the node attribute from the latent space can be achieved following existing designs of autoencoders (Zhai et al., 2018). However, decoding the edges connected to a node from the latent space in the entire graph can be computationally expensive, considering that a node can potentially connect to any other nodes within the graph as its neighbors. To address this challenge, we propose a Bi-Level Neighborhood Decoder (BLND), which decodes each latent feature into an inter-cluster neighbor map and an intra-cluster neighbor map defined below.

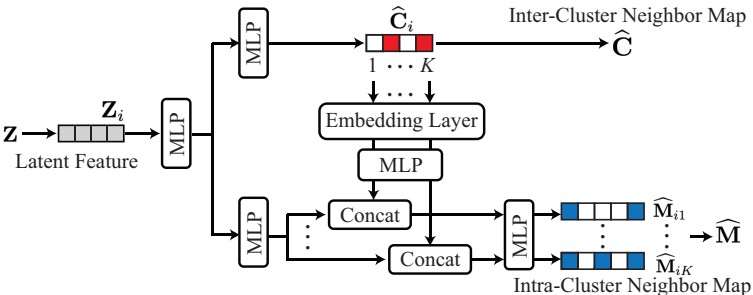

Figure 4: The structure of the Bi-Level Neighborhood Decoder (BLND), which generates an inter-cluster neighbor map and an intra-cluster neighbor map for each node.

**Preparing Inter-Cluster Neighbor Map and Intra-Cluster Neighbor Map.** We first divide the nodes in the graph into several clusters of similar size by applying balanced $K$-means clustering (Malinen & Fränti, 2014) on the node attributes. The number of clusters, denoted by $K$, is decided by cross-validation on each dataset. We set the maximal capacity of each cluster to $M = \lceil \frac{N}{K} \rceil$. After clustering the nodes into $K$ clusters, we assign an index to each node within its respective cluster by ordering them according to their indices in the entire graph. For each node $v_i$, the BLND first reconstructs an inter-cluster neighbor map which is denoted by a matrix $\mathbf{C} \in \{0, 1\}^{N \times K}$, where $\mathbf{C}_{ik} = 1$ if and only if there exists at least some node $v_j$ in cluster $k$ which is connected to node $v_i$. Our BLND then reconstructs the intra-cluster neighbor map between each node and each cluster, and the intra-cluster neighbor map is denoted by a three-dimensional tensor $\mathbf{M} \in \{0, 1\}^{N \times K \times M}$. $\mathbf{M}_{ikm} = 1$ if and only if the node $v_i$ is connected to the $m$-th node in cluster $k$. The detailed reconstruction processes for $\mathbf{C}$ and $\mathbf{M}$ are described in the following paragraph.

**Training the GAE with Bi-Level Neighborhood Decoder.** As illustrated in Figure 4, BLND reconstructs the inter-cluster neighbor map and intra-cluster neighbor map with two branches of MLP decoding layers. For each node $v_i$, BLND first reconstructs its inter-cluster neighbor map $\widehat{\mathbf{C}}_i$ with one MLP layer. After that, the predicted cluster indices $\mathcal{C}(i) = \{k \in [K] | \mathbf{C}_{ik} = 1\}$ are separately fed to an embedding layer to generate a set of class-conditional features $\mathcal{Z}(i) = \left\{ g(k) \in \mathbb{R}^{D'} | k \in \mathcal{C}(i) \right\}$ using the class-conditional embedding method in Classifier-free Guidance (Ho & Salimans, 2022), where $g$ contains one text embedding layer followed by an MLP layer. Next, each of the class-conditional features $g(k) \in \mathcal{Z}(i)$ is concatenated with the latent feature of the other branch for decoding the intra-cluster neighbor map by $\mathbf{M}_{ik} = g'(\mathbf{Z}_i \| g(k))$, where $g'$ is another MLP layer. The class-conditional embedding enables the parameter-sharing of the decoder layers for reconstructing intra-cluster neighbor maps of different clusters and thus largely reduces the computation cost for reconstructing the edges connected to a node in a large graph.

In our work, we train the GAE with BLND by minimizing the node reconstruction loss and the bi-level edge reconstruction loss

$$L_{\text{GAE}} = \underbrace{\left\| \mathbf{X} - \widehat{\mathbf{X}} \right\|_2^2}_{\text{Node Reconstruction Loss}} + \underbrace{\left( \left\| \mathbf{C} - \widehat{\mathbf{C}} \right\|_2^2 + \left\| \mathbf{M} - \widehat{\mathbf{M}} \right\|_2^2 \right)}_{\text{Bi-Level Edge Reconstruction Loss}} \tag{1}$$

where $\|\cdot\|_2$ denotes the Euclidean norm. The node reconstruction loss aims to reconstruct the node attribute. The bi-level edge reconstruction loss aims to reconstruct the inter-cluster neighbor map and the intra-cluster neighbor map. With the reconstructed inter-cluster neighbor map and the intra-cluster neighbor map, we can easily obtain the edges connected to the nodes in the graph.

## 4.2 Conditional Generation of Synthetic Graph Structures in Latent Space

After finishing the training of GAE, the node attributes and the edges connected to the nodes in the graph can be encoded into a continuous low-dimensional latent space. To generate synthetic graph structures, which are synthetic nodes and associated edges, we train a conditional Latent Diffusion Model (LDM) (Rombach et al., 2022) with Classifier-Free Guidance (CFG) (Ho & Salimans, 2022) on the latent features. The labels of synthetic nodes and Gaussian noises are fed to the LDM model to sample latent features of synthetic graph structures by Equation (17) in Section B of the appendix. With the conditional LDM trained on the

latent space, we can generate latent features of the synthetic graph structures. Next, we reconstruct the generated latent features into the node attributes and the edges associated with the nodes. Let $\mathcal{V}_L \subseteq \mathcal{V}$ denote the set of labeled nodes in the original graph $\mathcal{G}$, and $\mathcal{V}_{\text{syn}}$ denotes the set of generated synthetic nodes by DoG. Let the node attributes of $\mathcal{V}_{\text{syn}}$ generated by DoG be $\mathbf{X}_{\text{syn}}$. Let $C$ be the number of classes, and $N_0$ be the number of synthetic nodes for each class. $N' = CN_0 = |\mathcal{V}_{\text{syn}}|$ is the number of the synthetic nodes. Let $Y_{\text{syn}} = \{\widehat{y}_i\}_{i=1}^{N'}$ denote the set of the labels of the synthetic nodes. Let $\widehat{\mathbf{Z}} = \left\{\widehat{\mathbf{Z}}_i\right\}_{i=1}^{N'}$ denote the latent features of the synthetic graph structures. Let $\mathbf{A}_{\text{syn}} \in \mathbb{R}^{N' \times N}$ denote the edges of the synthetic nodes generated by our DoG. Then the augmented synthetic graph structure is $(\mathcal{V}_{\text{syn}}, \mathbf{X}_{\text{syn}}, \mathbf{A}_{\text{syn}})$. The adjacency matrix of the augmented graph is $\mathbf{A}_{\text{aug}} = [\mathbf{A}\ \mathbf{A}_{\text{syn}}; \mathbf{A}_{\text{syn}}\ \mathbf{A}] \in \mathbb{R}^{(N+N') \times (N+N')}$, and the node attributes of the augmented graph is $\mathbf{X}_{\text{aug}} = [\mathbf{X}; \mathbf{X}_{\text{syn}}] \in \mathbb{R}^{(N+N') \times D}$. The augmented graph, which is the combination of the original graph $\mathcal{G}$ and the synthetic graph structure, is then denoted by $\mathcal{G}_{\text{aug}} = (\mathcal{V} \cup \mathcal{V}_{\text{syn}}, \mathbf{X}_{\text{aug}}, \mathbf{A}_{\text{aug}})$. The set of labeled nodes in $\mathcal{G}_{\text{aug}}$ is $\mathcal{V}_L \cup \mathcal{V}_{\text{syn}}$ their labels are $Y_L \cup Y_{\text{syn}}$. Algorithm 1 and Algorithm 2 in Section D of the appendix describe the training algorithm of the DoG and the generation process of the augmented graph $\mathcal{G}_{\text{aug}}$ in details.

### 4.3 Low-Rank Regularization

After obtaining the augmented graph $\mathcal{G}_{\text{aug}} = (\mathcal{V} \cup \mathcal{V}_{\text{syn}}, \mathbf{X}_{\text{aug}}, \mathbf{A}_{\text{aug}})$ with $\bar{N} = N + N'$ nodes, we can train Graph Neural Networks (GNNs) for node classification on it. However, due to the randomness of the forward process in diffusion models (Ho et al., 2020) and the difficulty in accurately conditioning the LDM with class signals (Fu et al., 2024), the synthetic graph structures in the augmented graph inevitably contain noise, which can degrade the performance of GNNs trained on it. To address this issue, we propose a low-rank learning method with a low-rank regularization in the training of the GNNs on the augmented graph to enforce the learned feature kernel to be low-rank, whose motivation is discussed below.

**Motivation of Learning Low-Rank Features.** Let $\mathbf{Y} = [\mathbf{y}_1; \mathbf{y}_2; \dots; \mathbf{y}_{\bar{N}}] \in \{0, 1\}^{\bar{N} \times C}$ be the ground truth label matrix of all the nodes in the augmented graph, where $C$ is the number of classes. Let $\mathbf{K}$ be the gram matrix of the node representations $\mathbf{H} \in \mathbb{R}^{\bar{N} \times d}$, which is calculated by $\mathbf{K} = \mathbf{H}^\top \mathbf{H} \in \mathbb{R}^{\bar{N} \times \bar{N}}$. $\mathbf{H}$ can be the input to the last linear layer of the GNN trained for node classification or the output of the graph encoder in graph contrastive learning (GCL). By the low frequency property on the augmented graph illustrated in Figure 5 of Section C of the appendix, the projection of $\mathbf{Y}$ on the top $r$ eigenvectors of $\mathbf{K}$ with a small rank $r$, such as $r = 0.2N$, covers the majority of the information in $\mathbf{Y}$. This observation motivates using the low-rank part of the features $\mathbf{H}$, or equivalently the low-rank part of the gram matrix $\mathbf{K}$, for node classification. This is because the low-rank part of the features $\mathbf{H}$ covers the dominant information in the ground truth label matrix $\mathbf{Y}$. Through mostly using only the low-rank part of $\mathbf{H}$ for node classification, the negative affect of the noise introduced by the synthetic graph structures in the high-rank part of the features $\mathbf{H}$ on a trained GNN is reduced.

**Learning with Low-Rank Regularization.** Let $\left\{\widehat{\lambda}_i\right\}_{i=1}^{\bar{N}}$ with $\widehat{\lambda}_1 \geq \widehat{\lambda}_2 \dots \geq \widehat{\lambda}_{\min\{\bar{N},d\}} \geq \widehat{\lambda}_{\min\{\bar{N},d\}+1} = \dots, = 0$ be the eigenvalues of $\mathbf{K}$. In order to encourage the features $\mathbf{H}$ or the gram matrix $\mathbf{K}$ to be low-rank, we explicitly add the truncated nuclear norm $\|\mathbf{H}\|_{r_0+1} := \sum_{r=r_0}^{\bar{N}} \widehat{\lambda}_i$ to the training loss function of semi-supervised node classification. The starting rank $r_0 < \min(\bar{N}, d)$ is the rank of the features $\mathbf{H}$ we aim to keep in the node representation, that is, if $\|\mathbf{H}\|_{r_0} = 0$, then $\text{rank}(\mathbf{H}) = r_0$. To enforce the features $\mathbf{H}$ or the gram matrix $\mathbf{K}$ to be low-rank in the training of the GNNs, we regularize the training by adding a low-rank regularization term, which is the truncated nuclear norm $\|\mathbf{H}\|_{r_0}$, to the usual cross-entropy loss:

$$\min_{\mathbf{W}} L(\mathbf{W}) = \frac{1}{m} \sum_{v_i \in \mathcal{V}_L} \text{KL}\left(\mathbf{y}_i, [\text{softmax}\,(\mathbf{H}\mathbf{W})]_i\right) + \tau \|\mathbf{H}\|_{r_0}. \tag{2}$$

Here KL is the KL divergence between the label $\mathbf{y}_i$ and the softmax of the classifier output at node $v_i$. $\tau > 0$ is the weighting parameter for the truncated nuclear norm $\|\mathbf{H}\|_{r_0}$. We use a regular gradient descent to optimize (2) with a learning rate $\eta \in (0, \frac{1}{\widehat{\lambda}_1})$. The low-rank regularization can also be adapted to graph contrastive learning (GCL) methods by adding the truncated nuclear norm $\|\mathbf{H}\|_{r_0}$ to the training loss of the GCL methods. GCL methods usually optimize a contrastive loss, such as the InfoNCE loss (Hassani &

Ahmadi, 2020), to train a graph encoder that generates discriminative node representations $\mathbf{H}$. By jointly optimizing the truncated nuclear norm $\|\mathbf{H}\|_{r_0}$ with the contrastive loss of GCL methods, the learned node representations $\mathbf{H}$ can also be enforced to be low-rank. The rank $r_0$ is tuned by the standard cross-validation described in Section 5.1. DoG enhances the training of GNNs with a selected number of synthetic nodes, which is also decided by cross-validation as described in Section 5.1.

**Theoretical Justification for Low-Rank Regularization on the Augmented Graph.** The classifier in the optimization problem (2) is in fact a linear classifier with output $\mathbf{HW}$, and the predicted class scores are the softmax of such output. In Section A.1 of the appendix, we present the upper bound for the test loss of the linear classifier $\mathbf{HW}$ trained on the augmented graph under the standard setting for transductive learning. The test loss is the Euclidean distance between the output of the linear classifier, $\mathbf{HW}$, on the test nodes and the ground truth class label on the test nodes.

The upper bound for the test loss of the linear classifier is formally described in (4) of Section A.1 of the appendix. Such upper bound (4) theoretically justifies the usage of the low-rank regularization term $\|\mathbf{H}\|_{r_0}$ in the regularized training loss (2) for the linear classifier on the augmented graph, because a smaller $\|\mathbf{H}\|_{r_0}$ renders a lower upper bound for the test loss, benefiting the prediction accuracy for node classification using such linear classifier.

## 4.4 Complexity Analysis

In our work, we proposed a novel decoder, BLND, to reconstruct the edges connected to a node in the graph. To show its efficiency, we analyze the inference time complexity and the parameter size of the GAE with BLND. For comparison, we also analyze the inference time complexity and the parameter size of GAE, where BLND is replaced by a regular edge decoder that directly reconstructs the adjacency matrix $\mathbf{A}$. For ease of comparison, we denote the number of parameters and inference cost of all the MLP and GAT layers except BLND as $S_{\text{MLP}}$ and $C_{\text{MLP}}$ respectively. For a node $v_i$ in the graph, let $d_i = \sum_{k=1}^{K} \widehat{\mathbf{C}}_{ik}$ be the number of clusters predicted to be connected to $v_i$. Let $D'$ be the dimension of the input feature for BLND. The inference time complexity of GAE with BLND is $\mathcal{O}(KD' + d_i D'M + C_{\text{MLP}})$, where $\mathcal{O}(KD')$ is the additional complexity for computing the inter-cluster neighbor map and encoding the cluster indices. $\mathcal{O}(d_i D'M)$ is the computation cost for computing the intra-cluster neighbor map. In contrast, the inference time complexity of GAE with a regular edge decoder is $\mathcal{O}(D'KM + C_{\text{MLP}})$. We note that $d_i$ is upper bounded by the degree of the node $v_i$. In most graph datasets, the average degree of nodes is usually very small. For instance, on Pubmed, where the average node degree is 2.25, we have $d_i \leq 2.25$. As a result, $D'(K + d_i M) \ll D'KM$. For example, setting $K = 100$ and $M = 198$ on Pubmed, we find that the inference time complexity of GAE with BLND is $\mathcal{O}(545.5D' + C_{\text{MLP}})$, which is much more efficient than the regular edge decoder whose inference time complexity is $\mathcal{O}(19800D' + C_{\text{MLP}})$. In general, the inference time complexity of GAE with BLND is much lower than that of GAE with a regular edge decoder.

In addition, the parameter size of GAE with BLND is $D'^2 + D'K + D'M + S_{\text{MLP}}$, where $D'^2$ is the number of parameters in the layer for encoding cluster indices. $D'K$ is the number of parameters in the layer for predicting the inter-cluster neighbor map. $D'M$ is the number of parameters in the layer for reconstructing the intra-cluster neighbor map. It is noted that the reconstruction of the intra-cluster neighbor map of different clusters shares the same MLP decoder, as illustrated in Figure 4. In contrast, the parameter size of GAE with a regular edge decoder is $D'KM + S_{\text{MLP}}$. Because $K + M \ll KM$, the parameter size of GAE with BLND is much smaller than that of GAE with a regular edge decoder. For example, $KM = N = 19717$ on Pubmed with $M = 198$ and $K = 100$.

To thoroughly study the complexity of GAE with BLND, we also analyze its training time complexity. Let $t_{\text{train}}$ be the number of training epochs. The training time complexity of GAE with BLND should be $\mathcal{O}((KD' + KMD' + C_{\text{MLP}})Nt_{\text{train}})$. The time of training and data generation for both DoG and the variant of DoG, which replaces BLND with an MLP decoder reconstructing the entire neighbor map, is shown in Table 9 in Section E.4 of the appendix. DOG trains the GAE with BLND to encode node features and edges into low-dimensional latent features, which improves the efficiency of the diffusion process by reducing the dimension of input features. We remark that existing works on accelerating LDMs (Castells et al., 2024; Yang et al., 2023) can also be adapted to accelerate the diffusion process of DoG.

# 5 Experiments

In this section, we perform empirical evaluations of DoG on public graph benchmarks. In Section 5.1, we discuss the experimental setup and implementation details. In Section 5.2, we present the evaluation results of DoG for semi-supervised node classification and graph contrastive learning. Comprehensive ablation studies on DoG are performed in Section 5.3. Although the experiments are mostly conducted on homophilic graph datasets in this section, we also demonstrate the effectiveness of DoG on heterophilic graph datasets in Table 7 in Section E.2 of the appendix. More additional experiment results are deferred to Section E of the appendix. In Section E.3 of the appendix, we evaluate the quality of the generated synthetic graph structures. In Section E.4, we evaluate the training time and synthetic data generation time of DoG. In Section E.5, we list the cross-validation results of the hyper-parameters on different graph datasets. In Section E.6 of the appendix, we perform an ablation study on the synthetic graph structures generated by DoG. In Section E.7 of the appendix, we evaluate the performance of DoG on the large-scale graph.

## 5.1 Experimental Settings

**Implementation Details.** We conduct experiments on five public benchmarks that are widely used for node classification on attributed graphs, namely Cora, Citeseer, Pubmed (Sen et al., 2008a), Coauthor CS, and ogbn-arxiv (Hu et al., 2020). Details on the statistics of the dataset are deferred to Table 6 in Section E.1 of the appendix. To generate synthetic nodes, we first train the GAE on the input graph. The training of the GAE in DoG is divided into two phases. In the first phase, we only optimize the node reconstruction loss in $L_{\text{GAE}}$ for 1000 epochs. In the second phase, we optimize both the node reconstruction loss and the bi-level edge reconstruction loss in $L_{\text{GAE}}$ for another 1000 epochs. We use Adam optimizer with a learning rate of 0.001 for the training of the GAE. The weight decay is set to $1 \times 10^{-5}$. We keep track of the exponential moving average (EMA) of the model during the training with a decay factor of 0.995. We set the number of clusters, $K$, in BLND to 100 in our experiments.

We use the AdamW optimizer to optimize the LDM with a learning rate of 0.0002 and a weight decay factor of 0.0001. The same embedding layer as in Classifier-Free Guidance (CFG) (Ho & Salimans, 2022) is used to encode the class information, which is described in Section B of the appendix. The guidance strength of CFG is set to 0.5 in our experiments. A three-layer Multilayer Perceptron (MLP) is used as the denoising model in the LDM, whose hidden dimension is set to 512. We train LDM for 3000 epochs and keep track of the exponential moving average (EMA) of the model during the training with a decay factor of 0.995. We perform all the experiments in our paper on one NVIDIA Tesla A100 GPU. In Table 9 in Section E.4 of the appendix, we present the training time and data generation time of DoG.

**Tuning $r_0, \tau$, and $N'$ by Cross-Validation.** We tune the rank $r_0$, the weight for the truncated nuclear loss $\tau$, and the number of synthetic nodes $N'$ by standard cross-validation on each dataset with different models. Let $r_0 = \lceil \gamma \min \{N, d\} \rceil$ where $\gamma$ is the rank ratio. Let $N' = \beta |\mathcal{V}_L|$, where $\beta$ characterizes the amount of synthetic nodes. We select the values of $\gamma$, $\tau$, and $\beta$ by performing 5-fold cross-validation on 20% of the training data in each dataset. In addition, the models are trained for only 40% of the total epoch numbers in the cross-validation process. These strategies ensure the efficiency of the cross-validation process. The value of $\gamma$ is selected from $\{0.1, 0.2, 0.3, 0.4, 0.5, 0.6, 0.7, 0.8, 0.9\}$. The value of $\tau$ is selected from $\{0.05, 0.1, 0.15, 0.2, 0.25, 0.3, 0.35, 0.4, 0.45, 0.5\}$. The value of $\beta$ is selected from $\{1, 2, 3, 4, 5, 6, 7, 8, 9, 10\}$. The selected values of $r_0, \tau$, and $N'$ on each dataset for different models are shown in Table 10 in Section E.5 of the appendix. As shown in Table 11 in Section E.5 of the appendix, the time for the cross-validation process on different datasets is acceptable and does not largely increase the computation cost of the training process.

**Compared Methods.** In our experiments, we first compare our method with eight semi-supervised node classification methods, namely GCN (Kipf & Welling, 2017), GAT (Veličković et al., 2018), APPNP (Klicpera et al., 2019), S²GC (Zhu & Koniusz, 2020), SGFormer (Wu et al., 2023b), and EXPHORMER (Shirzad et al., 2023). In addition to semi-supervised node classification methods, we also apply DoG to graph contrastive learning (GCL) methods and compare them with eight state-of-the-art GCL methods, including DGI (Veličković et al., 2019), GMI (Peng et al., 2020), GCA (Zhu et al., 2021), MVGRL (Hassani & Ahmadi, 2020), GraphMAE (Hou et al., 2022), BGRL (Thakoor et al., 2022), SGCL (Sun et al., 2024), and GGD (Zheng et al., 2022).

## 5.2 Experimental Results

**Semi-supervised Node Classification.** To validate the effectiveness of Degree of Graph (DoG) in enhancing the performance of Graph Neural Networks (GNNs) for semi-supervised node classification, we applied DoG to augment the graph data in the training process of three distinct GNN architectures, including the vanilla GCN (Kipf & Welling, 2017) and EXPHORMER (Shirzad et al., 2023). We train each model on the augmented graphs following their corresponding original implementation details. We use the standard split following the baseline methods (Kipf & Welling, 2017; Shirzad et al., 2023) for all the nodes from the original graph. The synthesized nodes are added to the training set. We run all experiments 10 times and report the mean and standard deviation of the accuracy. Results in Table 1 show that models incorporating DoG significantly outperform their corresponding baseline models, achieving state-of-the-art performance across all datasets. In addition, the results in Table 1 show that the data augmentation with synthetic graph structures alone, which are denoted as DoG w/o low-rank, also brings significant performance improvements over the base models. For example, DoG w/o low-rank improves the performance of EXPHORMER by 1.4% on Citeseer. Combined with low-rank regularization, the improvement is further enhanced to 1.9%.

Table 1: Semi-supervised node classification performance on benchmark datasets. The best result is highlighted in bold, and the second-best result is underlined. This convention is followed by all the tables in this paper. The number of synthetic nodes added into different datasets is deferred to Table 10 in Section E.5 of the appendix.

| Methods | Cora | Citeseer | Pubmed | Coauthor CS | ogbn-arxiv |
|---|---|---|---|---|---|
| GCN (Kipf & Welling, 2017) | $81.7 \pm 0.4$ | $70.5 \pm 0.3$ | $79.4 \pm 0.4$ | $92.9 \pm 0.1$ | $71.7 \pm 0.3$ |
| GAT (Veličković et al., 2018) | $83.0 \pm 0.7$ | $72.5 \pm 0.7$ | $79.0 \pm 0.3$ | $93.6 \pm 0.7$ | $73.2 \pm 0.2$ |
| APPNP (Klicpera et al., 2019) | $83.3 \pm 0.5$ | $71.8 \pm 0.5$ | $80.1 \pm 0.2$ | $94.0 \pm 0.3$ | $70.3 \pm 0.3$ |
| S$^2$GC (Zhu & Koniusz, 2020) | $83.5 \pm 0.2$ | $73.6 \pm 0.5$ | $80.2 \pm 0.5$ | $93.8 \pm 0.1$ | $71.2 \pm 0.3$ |
| SGFormer (Wu et al., 2023b) | $84.5 \pm 0.8$ | $72.6 \pm 0.4$ | $80.3 \pm 0.6$ | $93.7 \pm 0.4$ | $72.6 \pm 0.1$ |
| EXPHORMER (Shirzad et al., 2023) | $84.1 \pm 0.8$ | $72.9 \pm 0.4$ | $83.2 \pm 0.6$ | $95.7 \pm 0.2$ | $72.4 \pm 0.3$ |
| DoG w/o low-rank (GCN) | $83.0 \pm 0.3$ ($\uparrow 1.3$) | $72.4 \pm 0.3$ ($\uparrow 1.9$) | $81.6 \pm 0.2$ ($\uparrow 2.2$) | $94.0 \pm 0.3$ ($\uparrow 1.1$) | $72.7 \pm 0.4$ ($\uparrow 1.0$) |
| DoG (GCN) | $84.0 \pm 0.3$ ($\uparrow 2.3$) | $73.6 \pm 0.4$ ($\uparrow 3.1$) | $82.8 \pm 0.3$ ($\uparrow 3.4$) | $94.5 \pm 0.2$ ($\uparrow 1.6$) | $73.1 \pm 0.3$ ($\uparrow 1.4$) |
| DoG w/o low-rank (EXPHORMER) | $\underline{85.1 \pm 0.5}$ ($\uparrow 1.0$) | $\underline{74.3 \pm 0.3}$ ($\uparrow 1.4$) | $\underline{84.3 \pm 0.3}$ ($\uparrow 1.1$) | $\underline{96.7 \pm 0.4}$ ($\uparrow 1.0$) | $\mathbf{73.4 \pm 0.5}$ ($\uparrow 1.0$) |
| DoG (EXPHORMER) | $\mathbf{85.7 \pm 0.4}$ ($\uparrow 1.6$) | $\mathbf{74.8 \pm 0.2}$ ($\uparrow 1.9$) | $\mathbf{84.6 \pm 0.4}$ ($\uparrow 1.4$) | $\mathbf{96.9 \pm 0.3}$ ($\uparrow 1.2$) | $\mathbf{73.4 \pm 0.3}$ ($\uparrow 1.0$) |

Table 2: Node classification performance with features learned from graph contrastive learning on benchmark datasets. The best results on each dataset are highlighted in bold.

| Methods | Cora | Citeseer | Pubmed | Coauthor CS | ogbn-arxiv |
|---|---|---|---|---|---|
| DGI (Veličković et al., 2019) | $81.7 \pm 0.6$ | $71.5 \pm 0.7$ | $77.3 \pm 0.6$ | $92.2 \pm 0.6$ | $65.1 \pm 0.4$ |
| GMI (Peng et al., 2020) | $82.7 \pm 0.2$ | $73.0 \pm 0.3$ | $80.1 \pm 0.2$ | $91.0 \pm 0.0$ | $69.6 \pm 0.3$ |
| GCA (Zhu et al., 2021) | $82.7 \pm 0.2$ | $73.0 \pm 0.3$ | $80.1 \pm 0.2$ | $91.0 \pm 0.0$ | $69.6 \pm 0.3$ |
| MVGRL (Hassani & Ahmadi, 2020) | $82.9 \pm 0.7$ | $72.6 \pm 0.7$ | $79.4 \pm 0.3$ | $92.1 \pm 0.1$ | $68.1 \pm 0.1$ |
| BGRL (Thakoor et al., 2022) | $80.5 \pm 0.5$ | $71.0 \pm 0.5$ | $79.5 \pm 0.5$ | $93.3 \pm 0.1$ | $71.6 \pm 0.2$ |
| SGCL (Sun et al., 2024) | $82.5 \pm 0.3$ | $71.9 \pm 0.5$ | $81.2 \pm 0.5$ | $93.3 \pm 0.2$ | $71.0 \pm 0.1$ |
| GraphMAE (Hou et al., 2022) | $83.4 \pm 0.5$ | $73.0 \pm 0.5$ | $81.9 \pm 0.5$ | $93.0 \pm 0.1$ | $71.8 \pm 0.2$ |
| GGD (Zheng et al., 2022) | $83.9 \pm 0.6$ | $73.0 \pm 0.6$ | $81.3 \pm 0.8$ | $92.7 \pm 0.4$ | $71.6 \pm 0.5$ |
| DoG (MVGRL) | $84.1 \pm 0.4$ ($\uparrow 1.2$) | $74.0 \pm 0.5$ ($\uparrow 1.4$) | $81.8 \pm 0.3$ ($\uparrow 2.4$) | $93.5 \pm 0.1$ ($\uparrow 1.4$) | $69.9 \pm 0.1$ ($\uparrow 1.8$) |
| DoG (GraphMAE) | $\underline{84.7 \pm 0.5}$ ($\uparrow 1.3$) | $\underline{74.3 \pm 0.9}$ ($\uparrow 1.3$) | $\mathbf{83.2 \pm 0.5}$ ($\uparrow 1.3$) | $\mathbf{94.2 \pm 0.1}$ ($\uparrow 1.2$) | $\underline{72.8 \pm 0.2}$ ($\uparrow 1.0$) |
| DoG (GGD) | $\mathbf{85.2 \pm 0.5}$ ($\uparrow 1.3$) | $\mathbf{74.4 \pm 0.5}$ ($\uparrow 1.4$) | $\underline{82.8 \pm 0.6}$ ($\uparrow 1.5$) | $\underline{93.9 \pm 0.3}$ ($\uparrow 1.2$) | $\mathbf{73.0 \pm 0.3}$ ($\uparrow 1.4$) |

**Graph Contrastive Learning.** To demonstrate the power of DoG in augmenting the graph data for node-level learning tasks, we also perform experiments for using DoG to perform data augmentation in the training of graph contrastive learning (GCL) methods. We adhere to the experimental settings described in previous works (Hassani & Ahmadi, 2020; Thakoor et al., 2022; Sun et al., 2024) on GCL. After finishing the GCL pre-training, the encoders of the GCL methods are used to extract the node representations of all the nodes. Next, we evaluate the quality of the node representations by training a logistic regression model on the training data of each dataset for node classification. We run all experiments 10 times and report the mean and standard deviation of the accuracy. Results in Table 2 demonstrate that DoG significantly improves the performance of GCL methods. For instance, DoG (MVGRL) improves the performance of

MVGRL by 2.4% on Pubmed. In addition, DoG (MVGRL) improves the performance of MVGRL by an average of 1.64% across the five benchmarks.

## 5.3 Ablation Study

**Ablation Study on the Low-Rank Regularization.** In this section, we study the effectiveness of low-rank regularization in DoG for node classification. Since low-rank regularization is applied to mitigate the noise in the synthetic graph structures generated by the diffusion models, we compare low-rank regularization with two other data selection methods, namely TopoFilter (Wu et al., 2020) and NGC (Wu et al., 2021), that are proposed for learning with noisy data. Although the data selection methods are not specifically designed for learning with graph data, they can be easily adapted to perform data selection on the synthetic nodes added to the training set in the training process. GCN is used as the base model for this ablation study. It is observed from the results in Table 3 that low-rank regularization significantly improves the performance of node classification on the augmented graphs generated by DoG. DoG outperforms its counterpart without low-rank regularization by an average of 1.44% in classification accuracy on the five graph datasets. Furthermore, DoG outperforms the best results of the two data selection methods by an average of 1.02% in classification accuracy on the five graph datasets, highlighting its advantages in mitigating the adverse effects of noise in synthetic graph structures over existing methods.

Table 3: Ablation Study on the Low-Rank Regularization in DoG and Comparison with Data Selection Methods. The best results on each dataset are highlighted in bold.

| Methods | Synthetic Graph Structures | Cora | Citeseer | Pubmed | Coauthor CS | ogbn-arxiv |
|---|---|---|---|---|---|---|
| GCN (Kipf & Welling, 2017) | No | $81.7 \pm 0.4$ | $70.5 \pm 0.3$ | $79.4 \pm 0.4$ | $92.9 \pm 0.1$ | $71.7 \pm 0.3$ |
| GCN with low-rank | No | $82.6 \pm 0.5$ (↑ 0.9) | $72.8 \pm 0.7$ (↑ 2.3) | $81.0 \pm 0.5$ (↑ 1.6) | $93.6 \pm 0.4$ (↑ 0.7) | $72.4 \pm 0.4$ (↑ 0.7) |
| DoG w/o low-rank (GCN) | Yes | $83.0 \pm 0.3$ (↑ 1.3) | $72.4 \pm 0.3$ (↑ 1.9) | $81.6 \pm 0.2$ (↑ 2.2) | $94.0 \pm 0.3$ (↑ 1.1) | $72.7 \pm 0.4$ (↑ 1.0) |
| DoG w/o low-rank (GCN) + TopoFilter (Wu et al., 2020) | Yes | $83.0 \pm 0.3$ (↑ 1.3) | $72.3 \pm 0.6$ (↑ 1.8) | $81.3 \pm 0.4$ (↑ 1.9) | $93.6 \pm 0.2$ (↑ 0.7) | $72.1 \pm 0.3$ (↑ 0.4) |
| DoG w/o low-rank (GCN) + NGC (Wu et al., 2021) | Yes | $82.9 \pm 0.2$ (↑ 1.2) | $72.1 \pm 0.4$ (↑ 1.6) | $81.2 \pm 0.3$ (↑ 1.8) | $93.5 \pm 0.2$ (↑ 0.6) | $72.1 \pm 0.6$ (↑ 0.4) |
| DoG (GCN) | Yes | $\mathbf{84.0 \pm 0.3}$ (↑ 2.3) | $\mathbf{73.6 \pm 0.4}$ (↑ 3.1) | $\mathbf{82.8 \pm 0.3}$ (↑ 3.4) | $\mathbf{94.5 \pm 0.2}$ (↑ 1.6) | $\mathbf{73.1 \pm 0.3}$ (↑ 1.4) |

In addition to studying the performance of low-rank regularization in mitigating noise in synthetic graph structures. We also perform an ablation study by applying low-rank regularization to vanilla GCN trained on the original graph. The results in Table 3 show that applying low-rank regularization vanilla GCN can boost the performance by an average of 1.44% in classification accuracy on the five graph datasets. This improvement can be further boosted to 2.24% after combining augmentation with synthetic graph structures and low-rank regularization, demonstrating the effectiveness of augmentation with synthetic graph structures in DoG.

**Comparison with Existing Node-Level Data Augmentation Methods.** DoG is among the first to generate synthetic graph structures that improve the prediction accuracy of the GNNs for node classification by enlarging the size of the training set. The augmentation method of DoG is orthogonal to existing node-level data augmentation methods, which modifies the graph structure, node features, or training labels of nodes in the graph. In this section, we compare DoG with existing node-level data augmentation methods, NODEDUP (Guo et al., 2024), FLAG (Kong et al., 2022), GraphMix (Verma et al., 2021), TADA (Lai et al., 2024), LGGD (Azad & Fang, 2024), and GeoMix (Zhao et al., 2024), which are surveyed in Section 2.2. In addition, we evaluate the performance of combining the DoG with the competing node-level data augmentation methods, where the node-level data augmentation methods are applied to the augmented graph generated by DoG. It is observed from Table 4 that DoG performs better than all the competing node-level data augmentation methods. In addition, combining DoG with these node-level data augmentation methods further improves their performance on all the datasets. For instance, combining GeoMix with DoG improves the performance of GeoMix by 1.1% in node classification accuracy on Pubmed.

**Ablation Study on the Balanced K-means Clustering in BLND.** To verify the effectiveness of the balanced K-means clustering algorithm used in BLND, we perform an ablation study by replacing the balanced K-means clustering algorithm with random partitioning in the Bi-Level Neighborhood Decoder (BLND). Instead of splitting the nodes into 100 clusters with K-means, the nodes are randomly split into 100 groups, where each group has less than $\lceil \frac{N}{100} \rceil$ nodes and $N$ is the number of nodes in the graph. We also conducted a sensitivity analysis on the value of $K$ in the balanced K-means clustering on Cora with

Table 4: Comparisons between DoG and existing node-level data augmentation methods. The values in the parentheses are the improvements over the baseline model EXPHORMER (Shirzad et al., 2023).

| Methods | Cora | Citeseer | Pubmed | Coauthor CS | ogbn-arxiv |
|---|---|---|---|---|---|
| EXPHORMER (Shirzad et al., 2023) | $84.1 \pm 0.8$ | $72.9 \pm 0.4$ | $83.2 \pm 0.6$ | $95.7 \pm 0.2$ | $72.4 \pm 0.3$ |
| DoG (EXPHORMER) | $85.7 \pm 0.4$ (↑1.6) | $74.8 \pm 0.2$ (↑1.9) | $84.6 \pm 0.4$ (↑1.4) | $96.9 \pm 0.3$ (↑1.2) | $73.4 \pm 0.3$ (↑1.0) |
| NODEDUP (EXPHORMER) (Guo et al., 2024) | $84.7 \pm 0.4$ (↑0.6) | $73.7 \pm 0.7$ (↑0.8) | $83.6 \pm 0.4$ (↑0.4) | $95.8 \pm 0.4$ (↑0.1) | $72.7 \pm 0.2$ (↑0.3) |
| NODEDUP + DoG (EXPHORMER) | $85.8 \pm 0.5$ (↑1.7) | $75.2 \pm 0.6$ (↑2.3) | $84.9 \pm 0.4$ (↑1.7) | $97.0 \pm 0.6$ (↑1.3) | $73.5 \pm 0.4$ (↑1.1) |
| FLAG (EXPHORMER) (Kong et al., 2022) | $85.3 \pm 0.3$ (↑1.2) | $74.4 \pm 0.3$ (↑1.3) | $83.8 \pm 0.5$ (↑0.6) | $96.0 \pm 0.5$ (↑0.3) | $72.7 \pm 0.3$ (↑0.3) |
| FLAG + DoG (EXPHORMER) | $85.8 \pm 0.5$ (↑1.7) | $\underline{75.4 \pm 0.5}$ (↑2.5) | $85.0 \pm 0.3$ (↑1.8) | $97.0 \pm 0.5$ (↑1.3) | $\underline{73.7 \pm 0.5}$ (↑1.3) |
| GraphMix (EXPHORMER) (Verma et al., 2021) | $85.2 \pm 0.3$ (↑1.1) | $74.2 \pm 0.3$ (↑1.3) | $83.9 \pm 0.5$ (↑0.7) | $96.1 \pm 0.5$ (↑0.4) | $72.8 \pm 0.3$ (↑0.4) |
| GraphMix + DoG (EXPHORMER) | $86.1 \pm 0.5$ (↑2.0) | $75.2 \pm 0.5$ (↑2.3) | $85.0 \pm 0.4$ (↑1.8) | $\underline{97.1 \pm 0.4}$ (↑1.4) | $73.6 \pm 0.5$ (↑1.2) |
| TADA (EXPHORMER) (Lai et al., 2024) | $85.4 \pm 0.3$ (↑1.3) | $74.4 \pm 0.4$ (↑1.5) | $84.0 \pm 0.4$ (↑0.8) | $96.3 \pm 0.4$ (↑0.6) | $72.6 \pm 0.3$ (↑0.2) |
| TADA + DoG (EXPHORMER) | $\mathbf{86.5 \pm 0.5}$ (↑2.4) | $\mathbf{75.5 \pm 0.4}$ (↑2.6) | $\underline{85.1 \pm 0.5}$ (↑1.9) | $97.0 \pm 0.3$ (↑1.3) | $73.5 \pm 0.3$ (↑1.1) |
| LGGD (EXPHORMER) (Azad & Fang, 2024) | $85.5 \pm 0.4$ (↑1.4) | $74.2 \pm 0.3$ (↑1.3) | $83.9 \pm 0.3$ (↑0.7) | $96.2 \pm 0.3$ (↑0.5) | $72.7 \pm 0.3$ (↑0.3) |
| LGGD + DoG (EXPHORMER) | $86.2 \pm 0.5$ (↑2.1) | $75.1 \pm 0.5$ (↑2.2) | $84.8 \pm 0.4$ (↑1.6) | $96.8 \pm 0.4$ (↑1.1) | $\underline{73.7 \pm 0.5}$ (↑1.3) |
| GeoMix (EXPHORMER) (Zhao et al., 2024) | $85.0 \pm 0.3$ (↑0.9) | $74.2 \pm 0.3$ (↑1.3) | $84.1 \pm 0.4$ (↑0.9) | $96.4 \pm 0.4$ (↑0.7) | $72.7 \pm 0.3$ (↑0.3) |
| GeoMix + DoG (EXPHORMER) | $\underline{86.3 \pm 0.5}$ (↑2.2) | $75.3 \pm 0.5$ (↑2.4) | $\mathbf{85.2 \pm 0.5}$ (↑2.0) | $\mathbf{97.2 \pm 0.5}$ (↑1.5) | $\mathbf{73.8 \pm 0.3}$ (↑1.4) |

GCN as the base model. The results in Table 5 demonstrate that balanced K-means clustering significantly outperforms random partitioning, Furthermore, the performance of DoG with BLND is not sensitive to $K$.

Table 5: Ablation Study on Balanced K-means Clustering in Bi-Level Neighborhood Decoder (BLND).

| Cluster Number $K$ | 10 | 25 | 50 | 100 | 150 | 200 | 100 (Random Partition) |
|---|---|---|---|---|---|---|---|
| Cora ACC | **84.1** | 83.9 | $\underline{84.0}$ | $\underline{84.0}$ | 83.9 | 83.8 | 82.8 |

In addition, we present the evaluation results of DoG on heterophilic graph datasets in Table 7 in Section E.2 of the appendix. We evaluate the quality of the generated synthetic graph structures on different graph datasets in Table 8 in Section E.3 of the appendix. The training time and synthetic data generation time of DoG are presented in Table 9 in Section E.4 of the appendix. The ablation study in Table 12 in Section E.6 of the appendix demonstrates the effectiveness of the synthetic graph structures by comparing with ablation models, which also increase the number of nodes in the graph. Table 13 in Section E.7 of the appendix shows that DoG is also effective in augmenting large-scale graphs for semi-supervised node classification.

# 6    Conclusions

In this work, we propose a novel graph diffusion model termed Diffusion on Graph (DoG). DoG is designed to generate synthetic graph structures, which consist of the synthetic nodes and the synthetic edges connecting to the synthetic nodes. The synthetic graph structures generated by DoG can then be combined with the original graph to form an augmented graph for the training of node-level learning tasks, including node classification and graph contrastive learning. In addition, a Bi-Level Neighbor Map Decoder (BLND) is introduced in DoG to improve the efficiency of the generation process. To mitigate the adverse effect of the noise introduced by the synthetic graph structures, a provable low-rank regularization method is proposed. Extensive experiments on various graph datasets for both node classification and graph contrastive learning have been performed to demonstrate the effectiveness of DoG.

**Acknowledgments**

This material is based upon work supported by the U.S. Department of Homeland Security under Grant Award Number 17STQAC00001-07-00. The views and conclusions contained in this document are those of the authors and should not be interpreted as necessarily representing the official policies, either expressed or implied, of the U.S. Department of Homeland Security. This work is also partially supported by the 2023 Mayo Clinic and Arizona State University Alliance for Health Care Collaborative Research Seed Grant Program.

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

# A    Theoretical Results

Our low-rank regularization is theoretically inspired by Theorem A.1 in this section. We consider the semi-supervised or transductive node classification task on the original graph formulated in Section 3.1. The results can be straightforwardly applied to the augmented graph with the original graph replaced by the augmented graph.

Following the standard setting in transductive learning, we suppose that a set $\mathcal{V}_L \subseteq \mathcal{V}$ with $|\mathcal{V}_L| = m$ is sampled uniformly without replacement from $\mathcal{V}$ as the labeled training nodes, and the remaining nodes $\mathcal{V}_U = \mathcal{V} \setminus \mathcal{V}_L$ are the test nodes. Let $\tilde{\mathbf{Y}} = [\mathbf{y}_1; \mathbf{y}_2; \ldots; \mathbf{y}_N] \in \{0,1\}^{N \times C}$ be the ground truth label matrix of all the nodes in the original graph. Using the features $\tilde{\mathbf{H}} \in \mathbb{R}^{N \times d}$ on the original graph generated by the methods describe in Section 4.3, the optimization problem of semi-supervised or transductive node classification is

$$\min_{\mathbf{W} \in \mathbb{R}^{d \times C}} \tilde{L}(\mathbf{W}) = \frac{1}{m} \sum_{v_i \in \mathcal{V}_L} \mathrm{KL}\left(\mathbf{y}_i, \left[\mathrm{softmax}\left(\tilde{\mathbf{H}}\mathbf{W}\right)\right]_i\right).$$

For a matrix $\mathbf{A}$ with $N$ rows, we use $[\mathbf{A}]_{\mathcal{V}_U}$ to denote the submatrix of $\mathbf{A}$ formed by rows of $\mathbf{A}$ corresponding to the test nodes, that is, all the rows $\mathbf{A}_i$ with $v_i \in \mathcal{V}_U$ form the submatrix $[\mathbf{A}]_{\mathcal{V}_U}$. $[\mathbf{A}]_{\mathcal{V}_L}$ is defined in the same way. For the gram matrix $\tilde{\mathbf{K}} = \tilde{\mathbf{H}}\tilde{\mathbf{H}}^\top$ with eigenvalues $\tilde{\lambda}_1 \geq \tilde{\lambda}_2 \ldots \geq \tilde{\lambda}_N$, we denote by $\tilde{\mathbf{K}}_{\mathcal{V}_L, \mathcal{V}_L} := [\tilde{\mathbf{H}}]_{\mathcal{V}_L} [\tilde{\mathbf{H}}]_{\mathcal{V}_L}^\top$ the submatrix of $\tilde{\mathbf{K}}$ formed by rows and columns of $\mathbf{K}$ corresponding to nodes in $\mathcal{V}_L$.

We define $\tilde{\mathbf{F}}(\mathbf{W}, t) := \tilde{\mathbf{H}}\mathbf{W}^{(t)}$ as the output of the classifier after the $t$-th iteration of gradient descent on $\tilde{L}(\mathbf{W})$ for $t \geq 1$. We have the following theoretical result on the loss of the unlabeled test nodes $\mathcal{V}_\mathcal{U}$ measured by the gap between $\tilde{\mathbf{F}}(\mathbf{W}, t)$ and $\tilde{\mathbf{Y}}$.

**Theorem A.1.** Let $m \geq cN$ for a constant $c \in (0,1)$, and $r_0 \in [N]$. Assume that a set $\mathcal{V}_L \subseteq \mathcal{V}$ with $|\mathcal{V}_L| = m$ is sampled uniformly without replacement from $\mathcal{V}$ as the labeled training nodes, and the remaining nodes $\mathcal{V}_U = \mathcal{V} \setminus \mathcal{V}_L$ are the test nodes. Then for every $x > 0$, with probability at least $1 - \exp(-x)$, after

the $t$-th iteration of gradient descent on the training loss $\tilde{L}(\mathbf{W})$ for all $t \geq 1$, we have

$$\mathcal{U}_{\text{test}}(t) := \frac{1}{u}\left\|\left[\tilde{\mathbf{F}}(\mathbf{W}, t) - \tilde{\mathbf{Y}}\right]_{\mathcal{V}_U}\right\|_{\text{F}}^2$$

$$\leq \frac{c_0}{m}\left\|\left(\mathbf{I}_N - \eta\tilde{\mathbf{K}}_{\mathcal{V}_L, \mathcal{V}_L}\right)^t\left[\tilde{\mathbf{Y}}\right]_{\mathcal{V}_L}\right\|_{\text{F}}^2 + c_1 r_0 \left(\frac{1}{u} + \frac{1}{m}\right) + \left(\sqrt{\frac{\|\tilde{\mathbf{H}}\|_{r_0}}{u}} + \sqrt{\frac{\|\tilde{\mathbf{H}}\|_{r_0}}{m}}\right) + \frac{c_2 x}{u}, \quad (3)$$

where $\|\tilde{\mathbf{H}}\|_{r_0} := \sum_{r=r_0}^{N} \tilde{\lambda}_i$, $c_0, c_1, c_2$ are positive numbers depending on $\{\tilde{\lambda}_i\}_{i \in [N]}$ and $\tau_0$ with $\tau_0^2 = \max_{i \in [N]}[\hat{\mathbf{K}}]_{ii}$.

It is noted that $\mathcal{U}_{\text{test}}(t) = 1/u \cdot \left\|\left[\mathbf{F}(\mathbf{W}, t) - \tilde{\mathbf{Y}}\right]_{\mathcal{V}_U}\right\|_{\text{F}}^2$ is the test loss of the unlabeled nodes measured by the Euclidean distance between the classifier output $\mathbf{F}(\mathbf{W}, t)$ and the training label matrix $\tilde{\mathbf{Y}}$. We remark that the truncated nuclear norm $\|\tilde{\mathbf{H}}\|_{r_0}$ appears on the RHS of the upper bound (2).

## A.1 Upper Bound for the Test Loss of a GNN trained on the Augmented Graph

We now consider the bound for the test loss of a GNN trained on the augmented graph. Suppose that the labeled training nodes in the augmented graph $\bar{\mathcal{V}}_L$ are sampled uniformly without replacement from all the nodes of the augmented graph, $\mathcal{V} \cup \mathcal{V}_{\text{syn}}$, and the remaining nodes $\bar{\mathcal{V}}_U = [\bar{N}] \setminus \bar{\mathcal{V}}_L$ are the test nodes. By applying Theorem A.1 to the augmented graph and using features $\mathbf{H}$ in Section 4.3 to replace the features $\tilde{\mathbf{H}}$ in Theorem A.1, we conclude that the test loss of a GNN trained on the augmented graph is bounded by

$$\mathcal{U}_{\text{test}}(t) := \frac{1}{u}\left\|[\mathbf{F}(\mathbf{W}, t) - \mathbf{Y}]_{\bar{\mathcal{V}}_U}\right\|_{\text{F}} \leq$$

$$\frac{c_0}{m}\left\|\left(\mathbf{I}_N - \eta\mathbf{K}_{\bar{\mathcal{V}}_L, \bar{\mathcal{V}}_L}\right)^t[\mathbf{Y}]_{\bar{\mathcal{V}}_L}\right\|_{\text{F}}^2 + c_1 r_0\left(\frac{1}{u} + \frac{1}{m}\right) + \left(\sqrt{\frac{\|\mathbf{H}\|_{r_0}}{u}} + \sqrt{\frac{\|\mathbf{H}\|_{r_0}}{m}}\right) + \frac{c_2 x}{u}, \quad (4)$$

where $\|\mathbf{H}\|_{r_0}$ is the truncated nuclear norm defined in Section 4.3, $c_0, c_1, c_2$ are positive numbers depending on $\{\hat{\lambda}_i\}_{i \in [\bar{N}]}$ and $\tau_0$ with $\tau_0^2 = \max_{i \in [\bar{N}]} \mathbf{K}_{ii}$. $\mathbf{K}_{\bar{\mathcal{V}}_L, \bar{\mathcal{V}}_L} := [\mathbf{H}]_{\bar{\mathcal{V}}_L}[\mathbf{H}]_{\bar{\mathcal{V}}_L}^{\top}$ is the submatrix of $\mathbf{K}$ defined in Section 4.3 formed by rows and columns of $\mathbf{K}$ corresponding to nodes in $\bar{\mathcal{V}}_L$.

(4) theoretically justifies the usage of the truncated nuclear norm $\|\mathbf{H}\|_{r_0}$ in the regularized training loss (2) for GNNs on the augmented graph, as a smaller $\|\mathbf{H}\|_{r_0}$ leads to a tighter bound for the test loss. Moreover, when the low frequency property holds, which is usually the case as demonstrated by Figure 5 in Section C of the appendix, $\left\|\left(\mathbf{I}_N - \eta\mathbf{K}_{\bar{\mathcal{V}}_L, \bar{\mathcal{V}}_L}\right)^t[\mathbf{Y}]_{\mathcal{V}_L}\right\|_{\text{F}}$ would be very small with enough iteration number $t$.

## A.2 Proof of Theorem A.1

We present the proof of Theorem A.1 below.

*Proof.* It can be verified that at the $t$-th iteration of gradient descent for $t \geq 1$, we have

$$\mathbf{W}^{(t)} = \mathbf{W}^{(t-1)} - \eta\left[\tilde{\mathbf{H}}\right]_{\mathcal{V}_L}^{\top}\left[\tilde{\mathbf{H}}\mathbf{W}^{(t-1)} - \tilde{\mathbf{Y}}\right]_{\mathcal{V}_L}. \quad (5)$$

It follows by (5) that

$$\left[\tilde{\mathbf{H}}\right]_{\mathcal{V}_L}\mathbf{W}^{(t)} = \left[\tilde{\mathbf{H}}\right]_{\mathcal{V}_L}\mathbf{W}^{(t-1)} - \eta\left[\tilde{\mathbf{H}}\right]_{\mathcal{V}_L}\left[\tilde{\mathbf{H}}\right]_{\mathcal{V}_L}^{\top}\left[\tilde{\mathbf{H}}\mathbf{W}^{(t-1)} - \tilde{\mathbf{Y}}\right]_{\mathcal{V}_L}$$

$$= \left[\tilde{\mathbf{H}}\right]_{\mathcal{V}_L}\mathbf{W}^{(t-1)} - \eta\tilde{\mathbf{K}}_{\mathcal{V}_L, \mathcal{V}_L}^{\top}\left[\tilde{\mathbf{H}}\mathbf{W}^{(t-1)} - \tilde{\mathbf{Y}}\right]_{\mathcal{V}_L}, \quad (6)$$

With $\mathbf{F}(\mathbf{W}, t) = \mathbf{H}\mathbf{W}^{(t)}$, it follows by (6) that

$$\left[\mathbf{F}(\mathbf{W}, t) - \tilde{\mathbf{Y}}\right]_{\mathcal{V}_L} = \left(\mathbf{I}_m - \eta\tilde{\mathbf{K}}_{\mathcal{V}_L, \mathcal{V}_L}\right)\left[\mathbf{F}(\mathbf{W}, t - 1) - \tilde{\mathbf{Y}}\right]_{\mathcal{V}_L}.$$

It follows from the above equality that

$$\left[\mathbf{F}(\mathbf{W}, t) - \tilde{\mathbf{Y}}\right]_{\mathcal{V}_L} = \left(\mathbf{I}_N - \eta\tilde{\mathbf{K}}_{\mathcal{V}_L, \mathcal{V}_L}\right)^t\left[\mathbf{F}(\mathbf{W}, 0) - \tilde{\mathbf{Y}}\right]_{\mathcal{V}_L} = -\left(\mathbf{I}_N - \eta\tilde{\mathbf{K}}_{\mathcal{V}_L, \mathcal{V}_L}\right)^t\left[\tilde{\mathbf{Y}}\right]_{\mathcal{V}_L}, \qquad (7)$$

so that As a result of (7), we have

$$\left\|\left[\mathbf{F}(\mathbf{W}, t) - \tilde{\mathbf{Y}}\right]_{\mathcal{V}_L}\right\|_{\mathrm{F}} \leq \left\|\left(\mathbf{I}_N - \eta\tilde{\mathbf{K}}_{\mathcal{V}_L, \mathcal{V}_L}\right)^t\left[\tilde{\mathbf{Y}}\right]_{\mathcal{V}_L}\right\|_{\mathrm{F}}. \qquad (8)$$

We apply [Corollary 3.7] (Yang, 2023) to obtain the following bound for the test loss $\frac{1}{u}\left\|\left[\mathbf{F}(\mathbf{W}, t) - \mathbf{Y}\right]_{\mathcal{L}_U}\right\|_{\mathrm{F}}^2$:

$$\frac{1}{u}\left\|\left[\mathbf{F}(\mathbf{W}, t) - \tilde{\mathbf{Y}}\right]_{\mathcal{L}_U}\right\|_{\mathrm{F}}^2 \leq \frac{c_0}{m}\left\|\left[\mathbf{F}(\mathbf{W}, t) - \tilde{\mathbf{Y}}\right]_{\mathcal{V}_L}\right\|_{\mathrm{F}}^2 + c_1 \min_{0 \leq Q \leq n} r(u, m, Q) + \frac{c_2 x}{u}, \qquad (9)$$

with

$$r(u, m, Q) := Q\left(\frac{1}{u} + \frac{1}{m}\right) + \left(\sqrt{\frac{\sum\limits_{q=Q+1}^{n} \tilde{\lambda}_q}{u}} + \sqrt{\frac{\sum\limits_{q=Q+1}^{n} \tilde{\lambda}_q}{m}}\right),$$

where $c_0, c_1, c_2$ are positive numbers depending on $\{\tilde{\lambda}_i\}_{i \in [N]}$ and $\tau_0$ with $\tau_0^2 = \max_{i \in [N]} \left[\tilde{\mathbf{K}}\right]_{ii}$.

It follows by (7) and (9) with $Q = r_0$ that

$$\frac{1}{u}\left\|\left[\mathbf{F}(\mathbf{W}, t) - \tilde{\mathbf{Y}}\right]_{\mathcal{L}_U}\right\|_{\mathrm{F}}^2$$

$$\leq \frac{c_0}{m}\left\|\left(\mathbf{I}_N - \eta\tilde{\mathbf{K}}_{\mathcal{V}_L, \mathcal{V}_L}\right)^t\left[\tilde{\mathbf{Y}}\right]_{\mathcal{V}_L}\right\|_{\mathrm{F}}^2 + c_1 r_0\left(\frac{1}{u} + \frac{1}{m}\right) + \left(\sqrt{\frac{\sum\limits_{q=r_0+1}^{n} \tilde{\lambda}_q}{u}} + \sqrt{\frac{\sum\limits_{q=r_0+1}^{n} \tilde{\lambda}_q}{m}}\right) + \frac{c_2 x}{u}, \qquad (10)$$

which completes the proof using the notation $\left\|\tilde{\mathbf{H}}\right\|_{r_0} := \sum_{r=r_0}^{N} \tilde{\lambda}_i$.

$\square$

## B  Preliminaries of Diffusion Models

**Diffusion models (DMs)** (Sohl-Dickstein et al., 2015; Ho et al., 2020) are latent variable models that model the data $\mathbf{x}_0$ as Markov chains $\mathbf{x}_T \cdots \mathbf{x}_0$, with intermediate variables sharing the same dimension. DMs can be described with two Markovian processes: a forward diffusion process $q(\mathbf{x}_{1:T} \mid \mathbf{x}_0) = \prod_{t=1}^{T} q(\mathbf{x}_t \mid \mathbf{x}_{t-1})$ and a reverse denoising process $p_\theta(\mathbf{x}_{0:T}) = p(\mathbf{x}_T) \prod_{t=1}^{T} p_\theta(\mathbf{x}_{t-1} \mid \mathbf{x}_t)$. The forward process gradually adds Gaussian noise to data $\mathbf{x}_t$:

$$q(\mathbf{x}_t \mid \mathbf{x}_{t-1}) = \mathcal{N}(\mathbf{x}_t; \sqrt{1 - \beta_t}\mathbf{x}_{t-1}, \beta_t\mathbf{I}), \qquad (11)$$

where the hyperparameter $\beta_{1:T}$ controls the amount of noise added at each timestep $t$. The $\beta_{1:T}$ are chosen such that samples $\mathbf{x}_T$ can approximately converge to standard Gaussians, i.e., $q(\mathbf{x}_T) \approx \mathcal{N}(0, \mathbf{I})$. Typically, this forward process $q$ is predefined without trainable parameters.

The generation process of DMs is defined as learning a parameterized reverse denoising process, which aims to incrementally denoise the noisy variables $\mathbf{x}_{T:1}$ to approximate clean data $\mathbf{x}_0$ in the target data distribution:

$$p_\theta(\mathbf{x}_{t-1} \mid \mathbf{x}_t) = \mathcal{N}(\mathbf{x}_{t-1}; \boldsymbol{\mu}_\theta(\mathbf{x}_t, t), \rho_t^2 \mathbf{I}), \tag{12}$$

where the initial distribution $p(\mathbf{x}_T)$ is defined as $\mathcal{N}(0, \mathbf{I})$. The means $\boldsymbol{\mu}_\theta$ typically are neural networks such as U-Nets for images or Transformers for text, and the variances $\rho_t$ typically are also predefined.

As latent variable models, the forward process $q(\mathbf{x}_{1:T}|\mathbf{x}_0)$ can be viewed as a fixed posterior, to which the reverse process $p_\theta(\mathbf{x}_{0:T})$ is trained to maximize the variational lower bound of the likelihood of the data. However, directly optimizing the likelihood is known to suffer serious training instability (Nichol & Dhariwal, 2021). Instead, (Song & Ermon, 2019b; Ho et al., 2020) suggest a simple surrogate objective:

$$\mathcal{L}_{\mathrm{DM}} = \mathbb{E}_{\mathbf{x}_0, \boldsymbol{\varepsilon} \sim \mathcal{N}(0, \mathbf{I}), t}[||\boldsymbol{\varepsilon} - \boldsymbol{\varepsilon}_\theta(\mathbf{x}_t, t)||^2], \tag{13}$$

Intuitively, the model $\boldsymbol{\varepsilon}_\theta$ is trained to predict the noise vector $\boldsymbol{\varepsilon}$ to denoise diffused samples $\mathbf{X}_t$ at every step $t$ towards a cleaner one $\mathbf{X}_{t-1}$. After training, we can draw samples with $\boldsymbol{\varepsilon}_\theta$ by the iterative ancestral sampling:

$$\mathbf{X}_{t-1} = \frac{1}{\sqrt{1 - \beta_t}}(\mathbf{X}_t - \frac{\beta_t}{\sqrt{1 - \alpha_t^2}}\boldsymbol{\varepsilon}_\theta(\mathbf{X}_t, t)) + \rho_t \boldsymbol{\varepsilon}, \tag{14}$$

with $\boldsymbol{\varepsilon} \sim \mathcal{N}(\mathbf{0}, \mathbf{I})$. $\alpha_t$ and $\beta_t$ are constants with regard to the time $t$. The sampling chain is initialized from Gaussian prior $\mathbf{X}_T \sim p(\mathbf{X}_T) = \mathcal{N}(\mathbf{X}_T; \mathbf{0}, \mathbf{I})$.

**Latent Diffusion Models (LDMs)** (Rombach et al., 2022) extend the capabilities of standard Diffusion Models (DMs) by introducing a latent space representation that reduces the dimension of the data involved in the diffusion process. In LDMs, data $\mathbf{X}_0$ is first mapped to a lower-dimensional latent representation $\mathbf{Z}_0$, which undergoes the diffusion process instead of the high-dimensional original data. This mapping is defined by an encoder network $g_e(\mathbf{X}_0) = \mathbf{Z}_0$. The forward process in LDMs involves gradually adding Gaussian noise to the latent representations $\mathbf{Z}_t$:

$$q(\mathbf{Z}_t \mid \mathbf{Z}_{t-1}) = \mathcal{N}(\mathbf{Z}_t; \sqrt{1 - \beta_t}\mathbf{Z}_{t-1}, \beta_t \mathbf{I}), \tag{15}$$

where $\beta_{1:T}$ are hyperparameters controlling the noise addition, similar to DMs. The reverse process in LDMs aims to reconstruct the clean latent representation $\mathbf{Z}_0$ from the noisy representation $\mathbf{Z}_T$ by:

$$p_\theta(\mathbf{Z}_{t-1} \mid \mathbf{Z}_t) = \mathcal{N}(\mathbf{Z}_{t-1}; \boldsymbol{\mu}_\theta(\mathbf{Z}_t, t), \rho_t^2 \mathbf{I}), \tag{16}$$

with the reconstructed latent data $\mathbf{Z}_0$ subsequently transformed back to the original data space using a decoder network $g_d(\mathbf{Z}_0) = \mathbf{X}_0$. This lower-dimensional approach enhances computational efficiency and often improves the quality of the generated samples.

**Classifier-Free Guidance (CFG)** (Ho & Salimans, 2022) combines a conditional noise predictor $\boldsymbol{\varepsilon}_\theta(\mathbf{X}_t, \mathbf{C}, t)$ and an unconditional noise predictor $\boldsymbol{\varepsilon}_\theta(\mathbf{X}_t, t)$ in the sampling process to improve sample quality and enforce class guidance. CFG can be easily incorporated into LDMs with the sampling process formulated as follows:

$$\mathbf{X}_{t-1} = \frac{1}{\sqrt{1 - \beta_t}}(\mathbf{X}_t - \frac{\beta_t}{\sqrt{1 - \alpha_t^2}}\tilde{\varepsilon}_t) + \rho_t \boldsymbol{\varepsilon}, \tag{17}$$

where $\mathbf{Z} \sim \mathcal{N}(\mathbf{0}, \mathbf{I})$. $\alpha_t$, $\beta_t$ and $\rho_t$ are constants with regard to the time $t$. $\tilde{\varepsilon}_t = (1+\omega)\boldsymbol{\varepsilon}_\theta(\mathbf{X}_t, \mathbf{C}, t) - \omega\boldsymbol{\varepsilon}_\theta(\mathbf{X}_t, t)$, and $\omega$ is the guidance factor.

## C  Low Frequency Property

In this section, we propose the **low frequency property**, which suggests that the low-rank projection of the ground truth clean labels possesses the majority of the information of the clean labels, and projection of the label noise is mostly uniform over all the eigenvectors of a kernel matrix used in classification. The eigen-projection and the signal concentration ratio on five graph datasets, including Cora, Citeseer, Pubmed, Coauthor-CS, and ogbn-arxiv, are illustrated in Figure 5.

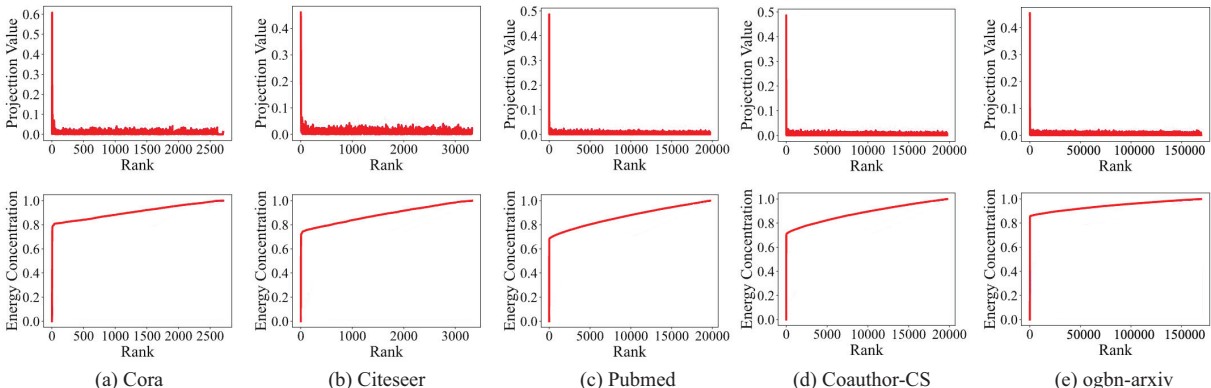

Figure 5: Eigen-projection (first row) and signal concentration ratio (second row) on the augmented graph for Cora, Citeseer, Pubmed, Coauthor-CS, and ogbn-arxiv. To compute the eigen-projection, we first calculate the eigenvectors $\mathbf{U}$ of the kernel gram matrix $\mathbf{K} \in \mathbb{R}^{\bar{N} \times \bar{N}}$ computed by a feature matrix $\mathbf{F} \in \mathbb{R}^{\bar{N} \times d}$, then the projection value is computed by $\mathbf{p} = \frac{1}{C} \sum_{c=1}^{C} \mathbf{U}^\top \mathbf{Y}^{(c)} / \left\| \mathbf{Y}^{(c)} \right\|_2^2 \in \mathbb{R}^n$, where $C$ is the number of classes, and $\mathbf{Y} \in \{0,1\}^{\bar{N} \times C}$ is the one-hot labels of all the training data in the augmented graph, $\mathbf{Y}^{(c)}$ is the $c$-th column of $\mathbf{Y}$. The eigen-projection $\mathbf{p}_r$ for $r \in [\min(\bar{N}, d)]$ reflects the amount of the signal projected onto the $r$-th eigenvector of $\mathbf{K}$, and the signal concentration ratio of a rank $r$ reflects the proportion of signal projected onto the top $r$ eigenvectors of $\mathbf{K}$. The signal concentration ratio for rank $r$ is computed by $\left\| \mathbf{p}^{(1:r)} \right\|_2$, where $\mathbf{p}^{(1:r)}$ contains the first $r$ elements of $\mathbf{p}$. For example, by the rank $r = 0.2 \min \left\{ \bar{N}, d \right\}$, the signal concentration ratio of $\mathbf{Y}$ for Cora, Citeseer, and Pubmed are 0.844, 0.809, 0.784, 0.779, and 0.787, respectively. We refer to such property as the **low frequency property**, which suggests that we can learn a low-rank portion of the observed label $\mathbf{Y}$, which covers most information in the ground truth clean label while only learning a small portion of the label noise.

# D  Algorithms for the Training of DoG and the Generation of the Augmented Graph

The training process of the DoG consists of two steps, which are detailed in Algorithm 1. The first step, which is from Line 1 to Line 5 in Algorithm 1, describes the training of the GAE on the original graph. For each of the nodes in the original graph, the encoder of the GAE encodes the node attributes of the given node, the node attributes of its neighbors, and the binary neighbor maps of edges connecting to that node into the latent representations. The positional embeddings of the given node and its neighbors are added to their node attributes to encode global positional information. The decoder of the GAE then reconstructs the node attributes of the given node and the binary neighbor maps of edges from the latent representations. The training of the GAE is supervised by the MSE loss. The second step, which is from Line 7 to Line 13, describes the training of the LDM on the latent representations of the nodes in the original graph following the standard training process as in (Rombach et al., 2022).

Algorithm 2 describes the generation process for the augmented graph $\mathcal{G}_{\mathrm{aug}}$, which also consists of two steps. In the first step, the conditional diffusion model generates a synthetic latent representation from a randomly sampled Gaussian noise given a node label. In the second step, the decoder of the GAE uses the synthetic latent representation to reconstruct the node attributes and the binary neighbor maps of edges connecting the synthetic node to authentic nodes in the original graph.

# E  Additional Experiments

## E.1  Datasets

In our experiment, we evaluate our method on five public benchmarks that are widely used for node representation learning, namely Cora, Citeseer, Pubmed (Sen et al., 2008a), Coauthor CS, and ogbn-arxiv (Hu

---

**Algorithm 1** Training DoG (Training the GAE and the LDM)

---

**Input:** The input attribute matrix $\mathbf{X}$, adjacency matrix $\mathbf{A}$, the training epochs of the GAE $t_{\text{GAE}}$, the labels $Y_L$ of the labeled nodes $\mathcal{V}_L$, and the training epochs of the LDM $t_{\text{LDM}}$
**Output:** The parameters of the GAE $\boldsymbol{\omega}$ and the parameters of the LDM $\boldsymbol{\theta}$
1: Obtain the inter-cluster neighbor map $\mathbf{C}$ and the intra-cluster neighbor map $\mathbf{M}$ by applying balanced $K$-Meamns clustering on $\mathbf{X}$
2: Initialize the parameter $\boldsymbol{\omega}$ of the GAE
3: **for** $t \leftarrow 1$ to $t_{\text{GAE}}$ **do**
4:     Update $\boldsymbol{\omega}$ by $\boldsymbol{\omega} \leftarrow \boldsymbol{\omega} - \eta \nabla_{\boldsymbol{\omega}} L_{\text{GAE}}$ with $L_{\text{GAE}}$ from Eq.(1)
5: **end for**
6: Initialize the parameter $\boldsymbol{\theta}$ of the LDM.
7: Map the node attributes $\mathbf{X}$ and the adjacency matrix $\mathbf{A}$ to the latent space using the encoder $g_e$ of the GAE as $\mathbf{H} = g_e(\mathbf{X}, \mathbf{A})$.
8: **for** $t \leftarrow 1$ to $t_{\text{LDM}}$ **do**
9:     Sample a Gaussian noise $\boldsymbol{\varepsilon} \sim \mathcal{N}(\mathbf{0}, \mathbf{I})$
10:     Get latent feature $\mathbf{Z}_L = \{\mathbf{Z}_i | v_i \in \mathcal{V}_L\}$ of $\mathcal{V}_L$
11:     Update $\boldsymbol{\theta}$ by $\boldsymbol{\theta} \leftarrow \boldsymbol{\theta} - \eta \nabla_{\boldsymbol{\theta}} \|\boldsymbol{\varepsilon}_{\boldsymbol{\theta}}(\mathbf{Z}_L, Y_L) - \boldsymbol{\varepsilon}\|_2^2$
12: **end for**
13: **return** The parameters of the GAE $\boldsymbol{\omega}$ and the parameters of the LDM $\boldsymbol{\theta}$

---

**Algorithm 2** Generation of the Augmented Graph $\mathcal{G}_{\text{aug}}$

---

**Input:** The input attribute matrix $\mathbf{X}$, adjacency matrix $\mathbf{A}$, the training epochs of the GNN $t_{\text{GNN}}$, the number of added nodes $M$, and the labels of the synthetic data $\{\widehat{y_i}\}_{i=1}^{N'}$
**Output:** The augmented graph $\mathcal{G}_{\text{aug}} = (\mathcal{V} \cup \mathcal{V}_{\text{syn}}, \mathbf{X}_{\text{aug}}, \mathbf{A}_{\text{aug}})$.
1: **for** $i \leftarrow 1$ to $M$ **do**
2:     Sample a Gaussian noise $\boldsymbol{\varepsilon} \sim \mathcal{N}(\mathbf{0}, \mathbf{I})$
3:     Set the class label of the $i$-th synthetic node to $\widehat{y_i}$
4:     Generate $\widehat{\mathbf{H}}_i$ from $\boldsymbol{\varepsilon}$ with the LDM for class $\widehat{y_i}$
5: **end for**
6: Decode $\widehat{\mathbf{Z}} = \{\widehat{\mathbf{Z}}_i\}_{i=1}^{N'}$ to $\mathbf{X}_{\text{syn}}$ and $\mathbf{A}_{\text{syn}}$ with the decoder of the GAE as $\mathbf{X}_{\text{syn}}, \mathbf{A}_{\text{syn}} = g_d(\widehat{\mathbf{Z}})$
7: Get $\mathbf{A}_{\text{aug}} = [\mathbf{A}\ \mathbf{A}_{\text{syn}}; \mathbf{A}_{\text{syn}}\ \mathbf{A}]$ and $\mathbf{X}_{\text{aug}} = [\mathbf{X}; \mathbf{X}_{\text{syn}}]$
8: **return** $\mathcal{G}_{\text{aug}} = (\mathcal{V} \cup \mathcal{V}_{\text{syn}}, \mathbf{X}_{\text{aug}}, \mathbf{A}_{\text{aug}})$

---

et al., 2020). Cora, Citeseer, and Pubmed are the three most widely used citation networks, where nodes are documents and edges are citation links. Coauthor CS is a co-authorship graph based on the Microsoft Academic Graph from the KDD Cup 2016 challenge. Nodes in Coauthor CS are authors that are connected by an edge if they co-authored a paper. Node features represent paper keywords for each author's papers, and class labels indicate the most active fields of study for each author. Nodes in the ogbn-arxiv dataset are publications from the arXiv with comprehensive node features derived from paper abstracts, which are used to predict the subject area of each paper. Edges in the ogbn-arxiv dataset represent citation relationships between papers. For all our experiments, we follow the default separation (Shchur et al., 2018; Mernyei & Cangea, 2020; Hu et al., 2020) of training, validation, and test sets on each benchmark.

Table 6: The statistics of the datasets.

| Dataset | Nodes | Edges | Features | Classes |
|---|---|---|---|---|
| Cora | 2,708 | 5,429 | 1,433 | 7 |
| CiteSeer | 3,327 | 4,732 | 3,703 | 6 |
| Pubmed | 19,717 | 44,338 | 500 | 3 |
| Coauthor CS | 18,333 | 81,894 | 6,805 | 15 |
| ogbn-arxiv | 169,343 | 1,166,243 | 128 | 40 |

It is worthwhile to mention that this work focuses on augmenting attributed graphs for node-level learning tasks. DoG generates synthetic node attributes and synthetic edges. DoG consists of a Graph Autoencoder (GAE) that encodes node attributes into latent node features. Subsequently, the latent diffusion model within DoG is trained on these latent features. Consequently, node attributes are essential for both the training and generation processes of the DoG model, so that DoG cannot be applied to non-attributed graphs.

## E.2 Evaluation on Heterophilic Graphs

In this section, we study the effectiveness of DoG for semi-supervised node classification on four widely used heterophilic graph datasets, namely Texas, Film, Chameleon, and Squirrel (Pei et al., 2020). We apply DoG to ACM-GCN (Luan et al., 2022), which is a widely used GNN for semi-supervised node classification on heterophilic graphs. Results in Table 7 show that DoG can also largely enhance the performance of GNN for semi-supervised node classification on heterophilic graph datasets. In addition, the results show that the data augmentation with synthetic graph structures alone, which are denoted as DoG w/o low-rank, also brings significant performance improvements over the base model. For example, DoG w/o low-rank

improves the performance of ACM-GCN by 1.4% on Squirrel. Combined with low-rank regularization, the improvement by DoG is further enhanced to 2.0%.

Table 7: Evaluation results of DoG on heterophilic graphs.

| Methods | Texas | Film | Chameleon | Squirrel |
|---|---|---|---|---|
| ACM-GCN (Pei et al., 2020) | $95.1 \pm 3.2$ | $41.6 \pm 1.1$ | $69.0 \pm 1.7$ | $58.0 \pm 1.9$ |
| DoG w/o low-rank (ACM-GCN) | $96.2 \pm 2.3$ ($\uparrow 1.1$) | $42.8 \pm 1.4$ ($\uparrow 1.2$) | $70.0 \pm 1.6$ ($\uparrow 1.0$) | $59.4 \pm 1.4$ ($\uparrow 1.4$) |
| DoG (ACM-GCN) | $96.6 \pm 2.1$ ($\uparrow 1.5$) | $43.2 \pm 1.2$ ($\uparrow 1.6$) | $70.6 \pm 1.4$ ($\uparrow 1.6$) | $60.0 \pm 1.3$ ($\uparrow 2.0$) |

### E.3   Study on the Quality of the Generated Synthetic Graph Structures

In this section, we study the quality of the generated synthetic graph structures by comparing two heuristic metrics on graph characteristics, which are the homophily ratio and the average node degree, between the original graph and generated synthetic graph structures. The homophily ratio, defined as the proportion of edges that connect nodes of the same class among all edges, verifies the structural integrity of synthetic edges in the generated synthetic graph structures compared to the edges in the original graph. The average node degree measures the connectivity pattern of nodes within a graph. By comparing the average node degree of the original graph with that of the synthetic graph structures, we can evaluate how well the connectivity patterns of the original graph are maintained in the synthetic graph structures. The results in Table 8 show that both the average node degree and homophily ratio of the generated synthetic graph structures closely match those of the original graph.

Table 8: The statistics of the datasets.

| Dataset | Homophily Ratio | | Average Node Degree | |
|---|---|---|---|---|
| | Original Graph | Synthetic Graph Structures | Original Graph | Synthetic Graph Structures |
| Cora | 0.81 | 0.82 | 4.0 | 3.8 |
| CiteSeer | 0.74 | 0.75 | 2.9 | 2.8 |
| Pubmed | 0.80 | 0.81 | 4.5 | 4.6 |
| Coauthor CS | 0.80 | 0.78 | 8.9 | 8.6 |
| ogbn-arxiv | 0.66 | 0.66 | 13.8 | 13.6 |

### E.4   Training Time and Synthetic Data Generation Time

To study the efficiency of BLND, we compare the training time between our GAE with BLND and GAE without BLND. In addition, we also compare the time for generation of our DoG and DoG without BLND in its GAE. All evaluations are conducted using a single Nvidia A100 GPU. It is observed from the results in Table 9 that BLND significantly reduces the computation cost of the training and synthetic graph structure generation. For instance, the training of GAE without BLND takes over five times the training time of our GAE with BLND on ogbn-arxiv. In addition, BLND also significantly reduces the time for synthetic graph structure generation. For instance, the data generation without BLND takes over four times the data generation time of our DoG with BLND on ogbn-arxiv.

Table 9: Time for the training of GAE and LDM in DoG and data generation with DoG on different datasets.

| Datasets | Training Time (minutes) | | | Generation Time (s/sample) | |
|---|---|---|---|---|---|
| | GAE | GAE without BLND | LDM | DoG | DoG without BLND |
| Cora | 11 | 16 | 39 | 0.049 | 0.066 |
| Citeseer | 14 | 18 | 41 | 0.052 | 0.067 |
| Pubmed | 41 | 129 | 154 | 0.067 | 0.073 |
| Coauthor CS | 52 | 145 | 179 | 0.074 | 0.088 |
| ogbn-arxiv | 301 | 1690 | 315 | 0.130 | 0.426 |

### E.5 Cross-Validation Results

**Tuning $r_0$, $\tau$, and $N'$ by Cross-Validation.** The selected values of $r_0$, $\tau$, and $N'$ on each dataset for different models are shown in Table 10. Since we set $r_0 = \lceil \gamma \min\{N, d\} \rceil$ and $N' = \beta|\mathcal{V}_L|$ in Section 5.1, we show the optimal values of $\gamma$, $\tau$, and $\beta$ in Table 10. Since we train different models for 40% of the total number of epochs on 20% of the training data, the cross-validation process does not largely increase the computation overhead for using DoG to augment the training of models for node classification and GCL.

Table 10: Selected rank ratio $\gamma$ and truncated nuclear loss's weight $\lambda$ for each dataset.

| Methods | Hyperparameters | Cora | Citeseer | Pubmed | Coauthor CS | ogbn-arxiv |
|---|---|---|---|---|---|---|
| DoG (GCN) | $\tau$ | 0.10 | 0.15 | 0.20 | 0.10 | 0.10 |
| | $\gamma$ | 0.2 | 0.2 | 0.2 | 0.2 | 0.2 |
| | $\beta$ | 3 | 3 | 2 | 1 | 1 |
| DoG (EXPHORMER) | $\tau$ | 0.10 | 0.10 | 0.20 | 0.10 | 0.15 |
| | $\gamma$ | 0.2 | 0.2 | 0.2 | 0.2 | 0.2 |
| | $\beta$ | 1 | 3 | 2 | 2 | 1 |
| DoG (MVGRL) | $\tau$ | 0.15 | 0.15 | 0.20 | 0.10 | 0.10 |
| | $\gamma$ | 0.2 | 0.2 | 0.2 | 0.2 | 0.2 |
| | $\beta$ | 2 | 2 | 2 | 1 | 1 |
| DoG (GraphMAE) | $\tau$ | 0.10 | 0.10 | 0.25 | 0.10 | 0.10 |
| | $\gamma$ | 0.2 | 0.2 | 0.2 | 0.2 | 0.2 |
| | $\beta$ | 2 | 3 | 1 | 2 | 1 |
| DoG (GGD) | $\tau$ | 0.15 | 0.10 | 0.25 | 0.10 | 0.05 |
| | $\gamma$ | 0.2 | 0.2 | 0.2 | 0.2 | 0.2 |
| | $\beta$ | 2 | 2 | 3 | 1 | 2 |

Table 11: Time for selecting optimal number synthetic nodes $N'$ with cross-validation.

| Datasets | Cross-validation Time (minutes) | | | | |
|---|---|---|---|---|---|
| | DoG (GCN) | DoG (EXPHORMER) | DoG (MVGRL) | DoG (GraphMAE) | DoG (GGD) |
| Cora | 2.5 | 10.5 | 13.5 | 16.5 | 1.5 |
| Citeseer | 2.0 | 11.4 | 15.4 | 18.4 | 1.4 |
| Pubmed | 12.4 | 39.5 | 52.5 | 58.5 | 10.5 |
| Coauthor CS | 14.7 | 42.5 | 57.5 | 63.5 | 11.5 |
| ogbn-arxiv | 37.1 | 90.5 | 104.5 | 117.5 | 33.5 |

### E.6 Ablation Study on the Synthetic Graph Structures

To evaluate the effectiveness of the synthetic graph structures generated by the DoG in improving the performance of GNNs for node classification, we compare the DoG with two categories of ablation models of the DoG, which do not incorporate the synthetic graph structures generated by the DoG. The first category of the ablation models, which is referred to as DoG-Duplicate, only adds the synthetic nodes generated by DoG to the training set of the original graph without connecting them to the real nodes in the original graph. The second category of ablation models, which is referred to as DoG-KNN, connects the synthetic nodes generated by the DoG to the real nodes in the original graph using the K-Nearest Neighbor (KNN) algorithm on the node attributes. The value of $K$ in the KNN algorithm is set to $K \in \{1, \lceil d_{avg}/4 \rceil, \lceil d_{avg}/2 \rceil, \lceil d_{avg} \rceil, \lceil 2 \times d_{avg} \rceil, \lceil 4 \times d_{avg} \rceil\}$, where $d_{avg}$ is the average node degree of the original real graph. EXPHORMER (Shirzad et al., 2023) is employed as the base model for semi-supervised node classification. The results are shown in Table 12. It is observed that although increasing the number of nodes can boost the performance of the base GNN model, DoG with the connectivity of the new nodes, or the synthetic nodes, significantly outperforms the ablation model, DoG-Duplicate, which does not have connectivity of the new nodes. For example, DoG outperforms DoG-Duplicate by 1.4% in the node classification accuracy on the Citeseer dataset. Furthermore, DoG exhibits much better performance than DoG-KNN with different $K$ values. For instance, the best-performing DoG-KNN model with $K = d_{avg}$ improves the classification accuracy of the base model from 72.9% to 73.8% on the Citeseer dataset, whereas the DoG model further elevates the classification accuracy to 74.8%. In addition, we compare DoG with NODEDUP (Guo et al., 2024), which enlarges the training set by duplicating real nodes with low degrees in the original graph. Following the settings in (Guo et al.,

2024), only the nodes with degrees not exceeding 2 are duplicated. Edges connecting the low-degree nodes and their own duplicates are created. It is observed that DoG also significantly outperforms NODEDUP. For instance, DoG outperforms NODEDUP by 1.1% in the node classification accuracy on the Citeseer dataset. The ablation study results underline the efficacy of the synthetic graph structures generated by DoG in boosting classification performance.

Table 12: Ablation study on the effectiveness of the synthetic graph structures generated by DoG.

| Methods | Cora | Citeseer | Pubmed | CoauthorCS | ogbn-arxiv |
|---|---|---|---|---|---|
| EXPHORMER | 84.1 | 72.9 | 83.2 | 95.7 | 72.4 |
| NODEDUP | 84.7 (↑ 0.6) | 73.7 (↑ 0.8) | 83.6 (↑ 0.4) | 95.8 (↑ 0.1) | 72.7 (↑ 0.3) |
| DoG-Duplicate | 84.3 (↑ 0.2) | 73.4 (↑ 0.5) | 83.5 (↑ 0.3) | 95.9 (↑ 0.2) | 72.4 (↑ 0.0) |
| DoG-KNN ($K = 1$) | 84.8 (↑ 0.7) | 73.7 (↑ 0.8) | 83.6 (↑ 0.4) | 96.2 (↑ 0.5) | 72.8 (↑ 0.4) |
| DoG-KNN ($K = \lceil d_{avg}/4 \rceil$) | 84.8 (↑ 0.7) | 73.6 (↑ 0.7) | 83.8 (↑ 0.6) | 96.2 (↑ 0.5) | 72.7 (↑ 0.3) |
| DoG-KNN ($K = \lceil d_{avg}/2 \rceil$) | 84.7 (↑ 0.6) | 73.7 (↑ 0.8) | 83.9 (↑ 0.7) | 96.3 (↑ 0.6) | 72.8 (↑ 0.4) |
| DoG-KNN ($K = \lceil d_{avg} \rceil$) | 84.9 (↑ 0.8) | 73.8 (↑ 0.9) | 83.9 (↑ 0.7) | 96.1 (↑ 0.4) | 72.6 (↑ 0.2) |
| DoG-KNN ($K = \lceil 2 \times d_{avg} \rceil$) | 84.5 (↑ 0.4) | 73.6 (↑ 0.7) | 84.0 (↑ 0.8) | 96.4 (↑ 0.7) | 72.6 (↑ 0.2) |
| DoG-KNN ($K = \lceil 4 \times d_{avg} \rceil$) | 84.4 (↑ 0.3) | 73.6 (↑ 0.7) | 83.8 (↑ 0.6) | 96.3 (↑ 0.6) | 72.7 (↑ 0.3) |
| DoG (EXPHORMER) | **85.7** (↑ 1.6) | **74.8** (↑ 1.9) | **84.6** (↑ 1.4) | **96.9** (↑ 1.2) | **73.4** (↑ 1.0) |

### E.7 Performance on Large-Scale Graph

To evaluate the performance of DoG on large-scale graphs, we conduct an experiment for semi-supervised node classification on AMiner-CS (Feng et al., 2020), which consists of more than half a million nodes (593,486 nodes) and 6,217,004 edges. We employ DoG to augment the original graph for the training of GRAND+ (Feng et al., 2022). The experiment is performed following the settings in Section 5.1. The results are shown in Table 13. It is observed that DoG significantly improves the performance of GRAND+ by 1.4% on AMiner-CS, demonstrating the scalability of DoG for synthetic graph structure generation on large-scale graphs.

Table 13: Node classification performance on AMiner-CS.

| Methods | GRAND+ | DoG (GRAND+) |
|---|---|---|
| Accuracy | 54.2% | **55.6%** |

