# OpenReview forum: "Diffusion on Graph: Augmentation of Graph Structure for Node Classification"
_TMLR — Accepted by TMLR_

### Review · Reviewer_6C6o · 2024-09-30

**Summary Of Contributions:**

This paper proposes Diffusion on Graph (DoG), which generates synthetic graph structures to boost the performance of GNNs. The synthetic graph structures generated by DoG are combined with the original graph to form an augmented graph for the training of node-level learning tasks, such as node classification and graph contrastive learning (GCL). To improve the efficiency of the generation process, a Bi-Level Neighbor Map Decoder (BLND) is introduced in DoG. To mitigate the adverse effect of the noise introduced by the synthetic graph structures, a low-rank regularization method is proposed for the training of graph neural networks (GNNs) on the augmented graphs. Source code of DoG is also released for reproducibility.

**Audience:**

Yes

**Claims And Evidence:**

Yes

**Requested Changes:**

Please see the above section for requested changes on improving the motivation of the writing, references, and experiments/datasets.

**Strengths And Weaknesses:**

While graph neural network (GNN) models are briefly mentioned in the Introduction section, they should be more comprehensively mentioned in the Related Work section as a subsection, since there are many recent GNNs that effectively operate on node classification and link generation tasks on both the transductive and inductive settings. The authors should reference the following works in their writing for relevance and recency:

[WWW 2019] Heterogeneous Graph Attention Network. In The World Wide Web Conference (WWW '19). Association for Computing Machinery, New York, NY, USA, 2022–2032. https://doi.org/10.1145/3308558.3313562

[IEEE ICDM 2021] "Bi-Level Attention Graph Neural Networks," in 2021 IEEE International Conference on Data Mining (ICDM), Auckland, New Zealand, 2021 pp. 1126-1131.
doi: 10.1109/ICDM51629.2021.00133

The authors also need to more clearly motivate why they are considering a graph diffusion model based approach instead other successful methods like GNNs, and graph variational autoencoder approaches and are also used to synthesize graphs.

Model architecture is well described and seems to be technically correct. Complexity analysis is also described in detail, which is helpful for evaluation. Model hyperparameters are provided and the experiment setup is described in sufficient detail.

While experiment results achieve good performance against recent baselines, there are some concerns that the authors should help address: It would help if the authors provided experiment dataset statistics like number of nodes, number of edges, type of links,  and also conduct experiments on other graph settings like attributed vs. non-attributed graphs, and a combination of small and large-scale graphs (million node/billion edge graphs).

---

> ### Author Response · Authors · 2024-10-16
> **Thank you for your review**
>
> We appreciate the review and the suggestions in this review. The raised issues are addressed below.
>
> **1 “While graph neural network (GNN) … mentioned in the Related Work section...The authors should reference the following works…”**
>
> We have added discussions on related works on Graph Neural Networks (GNNs) in Section 2.4 of the revised paper. The suggested references [1, 2] are also included.
>
> **2 “… why they are considering a graph diffusion model based approach instead other successful methods like GNNs, and graph variational autoencoder…”**
>
> Existing works using GNNs and Graph Variational Autoencoders (GVAEs) on graph-level synthetic graph generation [3, 4, 5, 6, 7] aim to generate a set of synthetic graphs resembling the graphs in the given graph datasets. In contrast, our work focuses on node-level synthetic graph structure generation, which aims to generate synthetic nodes and synthetic edges connecting the synthetic nodes to the original nodes in a given graph.  Since graph-level synthetic graph generation methods cannot generate synthetic nodes within a given graph, they cannot be applied to generate synthetic labeled nodes to enlarge the training set of a graph for node classification.
>
> **3 “…experiment dataset statistics like number of nodes, number of edges, type of links...”**
>
> The experiment dataset statistics including number of nodes, number of edges and type of links are provided in Table 6 in Section E.1 of the appendix.
>
> **4 ”...conduct experiments on other graph settings like attributed vs. non-attributed graphs...”**
>
> This work focuses on augmenting attributed graphs for node-level learning tasks. DoG generates synthetic node attributes and synthetic edges. DoG consists of a Graph Autoencoder (GAE) that encodes node attributes into latent node features. Subsequently, the latent diffusion model within DoG is trained on these latent features. Consequently, node attributes are essential for both the training and generation processes of the DoG model, so that DoG cannot be applied to non-attributed graphs.
>
> **5 “...a combination of small and large-scale graphs…”**
>
> To evaluate the performance of DoG on large-scale graphs, we conduct an experiment for semi-supervised node classification on AMiner-CS, which consists of more than half million nodes (593,486 nodes) and 6,217,004 edges. We employ DoG to augment the original graph for the training of GRAND+ [9]. The results are shown in the table below. It is observed that DoG significantly improves the performance of GRAND+ by $1.4\%$ on AMiner-CS, demonstrating the scalability of DoG for synthetic graph structure generation on large-scale graph.
>
> |   Methods    | Accuracy |
> | :----------: | :------: |
> |  GRAND+ [9]  | $54.2\%$ |
> | DoG (GRAND+) | $55.6\%$ |
>
> **References**
>
> [1] Wang, Xiao, et al. "Heterogeneous graph attention network." The world wide web conference. 2019.
>
> [2] Iyer, Roshni G., Wei Wang, and Yizhou Sun. "Bi-level attention graph neural networks." IEEE International Conference on Data Mining (ICDM), 2021.
>
> [3] Mitton, Joshua, et al. "A graph vae and graph transformer approach to generating molecular graphs." ICML 2020.
>
> [4] Bongini, P., M. Bianchini, and F. Scarselli. "Molecular graph generation with graph neural networks." stat (2021).
>
> [5] Zhu, Yanqiao, et al. "A survey on deep graph generation: Methods and applications." Learning on Graphs Conference. PMLR, 2022.
>
> [6] Vignac, Clement, et al. "Digress: Discrete denoising diffusion for graph generation." ICLR 2023.
>
> [7] Cai, Zhou, Xiyuan Wang, and Muhan Zhang. "Latent Graph Diffusion: A Unified Framework for Generation and Prediction on Graphs." arXiv preprint arXiv:2402.02518 (2024).
>
> [8] Tang, Jie, et al. "Arnetminer: extraction and mining of academic social networks." SIGKDD 2008.
>
> [9] Feng, Wenzheng, et al. "Grand+: Scalable graph random neural networks." Proceedings of the ACM Web Conference 2022.

---

### Review · Reviewer_qCN5 · 2024-10-24

**Summary Of Contributions:**

This paper investigate the graph generation problem within the context of network data, and aim to augment the original network with synthetic sub-graphs and boost the performance of GNNs.

The proposed method is based on diffusion, which was explored in the context of whole graph generation but has not been explored for generation of sub-graph structures in networks. Different from whole graph generation that generates one connected graph, the generation process described in this paper will generate multiple sub-structures distributed across different regions of the original graph.

The proposed model contains a Graph Autoencoder and a Latent Diffusion module. The encoder encodes a given node, as well as the connected edges, into latent space. The diffusion model then generates latent features of the synthetic graph structures. Then the decoder decodes the latent representations back into the data space. To match the generated structure to proper nodes in the graph, a bi-level neighborhood decoder is designed to reconstruct the edges connected to a node hierarchically. By training the model to reconstruct the given graph, the model learns the distribution of the graph, and can be used to generate new structures.

**Audience:**

Yes

**Broader Impact Concerns:**

No.

**Claims And Evidence:**

Yes

**Requested Changes:**

1. Please further clarify the motivation of leveraging latent diffusion for processing the latent representations.

2. One thing is unclear. The generated graph structure is generated based on a given node within the graph. Then it seems that the generated graph should be placed at the position of this node. Why do we have to find a position within the entire graph?

3. When feeding a given node with its edges into the model, which edges are included? Do the authors only consider the edges directly connected to the given node? If this is the case, it seems that the model can only process the information in a very local manner.

4. The high level introduction of the bi-level neighborhood decoder in the Introduction section should be improved. The current version cannot clearly explain how are the generated nodes and edges connected to the original graph.

**Strengths And Weaknesses:**

Strengths:

1. This paper proposes the first diffusion-based method to augment a given graph.

2. The proposed method obtains consistent performance improvement on different datasets.

Weakness:

1. Most baselines are not new.

2. Some parts are not explained clearly enough, as detailed below in the requested changes.

---

> ### Author Response · Authors · 2024-10-30
> **Response to Reviewer qCN5 Part 1**
>
> We appreciate the review and the suggestions in this review. The raised issues are addressed below.
>
> **Responses to the Weaknesses**
>
> **1. “Most baselines are not new.”**
>
> We have added three recently published node-level data augmentation methods to the revised paper, which are TADA [5], LGGD [6], and GeoMix [7], as the baselines for the DoG. The results are shown in the table below. It is observed that DoG performs better than all the competing node-level data augmentation methods. In addition, the augmentation method of DoG is orthogonal to the competing node-level data augmentation methods, which modifies the graph structure, node features, or training labels of nodes in the original graph. Combining DoG with these node-level data augmentation methods further improves their performance on all the datasets. For instance, combining GeoMix with DoG enhances the performance of GeoMix by $1.1\%$ in node classification accuracy on Pubmed.
>
> | Methods |Cora|Citeseer|Pubmed |Coauthor CS|ogbn-arxiv |
> | :-- | :--: | :--: | :--: | :--: | :--: |
> | EXPHORMER Exphormer| 84.1| 72.9 | 83.2| 95.7| 72.4 |
> | DoG (EXPHORMER)| 85.7 ($\uparrow$ 1.6)| 74.8 ($\uparrow$ 1.9) | 84.6 ($\uparrow$ 1.4)| 96.9 ($\uparrow$ 1.2)| 73.4 ($\uparrow$ 1.0)|
> | TADA (EXPHORMER) | 85.4 ($\uparrow$ 1.3)| 74.4 ($\uparrow$ 1.5) | 84.0 ($\uparrow$ 0.8)| 96.3 ($\uparrow$ 0.6)| 72.6 ($\uparrow$ 0.2) |
> | TADA + DoG (EXPHORMER) | **86.5** ($\uparrow$ 2.4)| **75.5** ($\uparrow$ 2.6) | 85.1 ($\uparrow$ 1.9)| 97.0 ($\uparrow$ 1.3)| 73.5 ($\uparrow$ 1.1) |
> | LGGD (EXPHORMER) | 85.5 ($\uparrow$ 1.4)| 74.2 ($\uparrow$ 1.3) | 83.9 ($\uparrow$ 0.7)| 96.2 ($\uparrow$ 0.5)| 72.7 ($\uparrow$ 0.3) |
> | LGGD + DoG (EXPHORMER) | 86.2 ($\uparrow$ 2.1)| 75.1 ($\uparrow$ 2.2) | 84.8 ($\uparrow$ 1.6)| 96.8 ($\uparrow$ 1.1)| 73.7 ($\uparrow$ 1.3) |
> | GeoMix (EXPHORMER) | 85.0 ($\uparrow$ 0.9)| 74.2 ($\uparrow$ 1.3) | 84.1 ($\uparrow$ 0.9)| 96.4 ($\uparrow$ 0.7)| 72.7 ($\uparrow$ 0.3) |
> | GeoMix + DoG (EXPHORMER) | 86.3 ($\uparrow$ 2.2)| 75.3 ($\uparrow$ 2.4) | **85.2** ($\uparrow$ 2.0)| **97.2** ($\uparrow$ 1.5)| **73.8** ($\uparrow$ 1.4) |
>
> **2.“Some parts are not explained clearly enough, as detailed below in the requested changes.”**
>
> Detailed responses to questions raised in the requested changes are below.
>
> **Responses to the Requested Changes**
>
> **1. “...the motivation of leveraging latent diffusion for processing the latent representations“**
>
> The motivation for leveraging the latent diffusion model (LDM) in processing the latent representations primarily stems from its ability to enhance the generative capabilities of the diffusion model while ensuring computational efficiency. The LDM trains the diffusion model in the compressed latent features generated by the graph autoencoder (GAE) rather than directly in the high-dimensional input data space. This approach significantly reduces the dimensionality of the data being processed, which decreases the computational resources required for the training and the inference of the diffusion model. For example, the dimension of input feature of Coauthor CS is 6805, and the GAE reduces it to 512, which significantly improves the computational efficiency for the training and the inference of the diffusion model.
> Moreover, the LDM has shown remarkable performance in generating high-quality synthetic data across various domains [1, 2]. By applying the LDM to the latent representations, the diffusion model captures the high-level patterns in the data more effectively, leading to improved performance on the generative tasks [1,2].

---

> > ### Author Response · Authors · 2024-10-30
> > **Response to Reviewer qCN5 Part 2**
> >
> > **2. “...The generated graph structure is generated based on a given node within the graph. Then it seems that the generated graph should be placed at the position of this node...”**
> >
> > To clarify the doubts about the generation process of synthetic graph structures, we first briefly summarize the training process and the generation process of the DoG below.
> >
> > The training process of the DoG consists of two steps, which are detailed in Algorithm 1 in Section D of the appendix. In the first step, the GAE is trained on the original graph. For each of the nodes in the original graph, the encoder of the GAE encodes the node attributes of the given node, the node attributes of its neighbors, and the binary neighbor maps of edges connecting to that node into the latent representations. The positional embeddings of the given node and its neighbors are added to their node attributes to encode global positional information.
> > The decoder of the GAE then reconstructs the node attributes of the given node and the binary neighbor maps of edges from the latent representations. The training of the GAE is supervised by the MSE loss.
> > In the second step, the conditional diffusion model is trained on the latent representations of the nodes in the original graph following the standard training process as in the LDM [1].
> >
> > The generation process of the DoG also consists of two steps, which are detailed in Algorithm 2 in Section D of the appendix. In the first step, the conditional diffusion model generates a synthetic latent representation from a randomly sampled Gaussian noise given a node label. In the second step, the decoder of the GAE uses the synthetic latent representation to reconstruct the node attributes and the binary neighbor maps of edges connecting the synthetic node to authentic nodes in the original graph.
> >
> > As described above, the DoG does not generate the synthetic graph structure, which is the synthetic node and edges connecting it to the given graph, based on a given node in the graph. Instead, the synthetic graph structure is generated from a Gaussian noise following the two-step generation process described above.
> >
> >
> > **3. ”When feeding a given node with its edges into the model, which edges are included? ...process the information in a very local manner.”**
> >
> > In the training process of the GAE in the DoG, only the edges directly connecting to a given node are included as the inputs to the encoder of the GAE. However, the positional embeddings of both the given node and its neighboring nodes are added to their node attributes to incorporate the global positional information. Existing works [3, 4] have shown that the positional embeddings of the nodes in the given graph can encode global positional information of nodes within the graph. Therefore, DoG utilizes both local and global information of the nodes in the given graph instead of only processing the information in a very local manner.
> >
> > **4. “... introduction of the bi-level neighborhood decoder in the Introduction section should be improved... how are the generated nodes and edges connected to the original graph.”**
> >
> > We have revised the introduction of the bi-level neighborhood decoder at the end of the third paragraph in the Introduction section to include an introduction on how the generated nodes and edges are connected to the original graph.
> >
> > The nodes in the original graph are first divided into several clusters using a balanced clustering method. In the generation process, the BLND first generates the binary inter-cluster neighbor map for each synthetic node, which identifies the clusters containing the nodes that the synthetic node should connect to. Subsequently, for each of the identified clusters, the BLND generates a binary intra-cluster neighbor map, which identifies the individual nodes within the cluster to which the synthetic node should connect.
> >
> > **References**
> >
> > [1] Rombach, Robin, et al. "High-resolution image synthesis with latent diffusion models." CVPR 2022.
> >
> > [2] Lovelace, Justin, et al. "Latent diffusion for language generation." NeurIPS 2023.
> >
> > [3] Ma, Liheng, Reihaneh Rabbany, and Adriana Romero-Soriano. "Graph attention networks with positional embeddings." PAKDD 2021.
> >
> > [4] You, Jiaxuan, Rex Ying, and Jure Leskovec. "Position-aware graph neural networks." ICML 2019.
> >
> > [5] Lai, Yurui, et al. "Efficient Topology-aware Data Augmentation for High-Degree Graph Neural Networks." KDD 2024.
> >
> > [6] Azad, Amitoz, and Yuan Fang. "A Learned Generalized Geodesic Distance Function-Based Approach for Node Feature Augmentation on Graphs." KDD 2024.
> >
> > [7] Zhao, Wentao, et al. "GeoMix: Towards Geometry-Aware Data Augmentation." KDD 2024.

---

### Review · Reviewer_su2i · 2024-10-27

**Summary Of Contributions:**

The authors propose a new node-level graph augmentation method outperforming current methods, by synthesizing new nodes and linking them to the original graph. They use latent diffusion to generate nodes and design a Bi-Level Neighborhood Decoder to reduce the time complexity when generating the adjacency matrix. The authors also add Low-Rank regularization during training to minimize the noise in the synthetic graph.

**Audience:**

Yes

**Claims And Evidence:**

Yes

**Requested Changes:**

•	The authors may analyze how the proposed method overcomes the challenges in labeling and imputing features and connectivity of new nodes when adding nodes to the original graph. (strengthen the work)
•	The authors could better justify whether the improvement in the proposed method comes from the increase in the number of nodes or the synthetic graph structures. (strengthen the work)

**Strengths And Weaknesses:**

Strong points：
•	The authors propose a graph augmentation method by generating synthetic nodes and edges to enlarge the training set and improve the performance of node classification models in the experiment.
•	The authors’ theoretical analysis and experiments prove the effectiveness of the proposed Bi-Level Neighborhood Decoder and Low-Rank regularization.

Weak points：
•	Adding a node or an edge can influence many other nodes, so it was once considered difficult to conduct data augmentation by adding nodes to the original graph [1,2]. In this paper, it is still unclear how the proposed method overcomes the challenges in labeling and imputing features and connectivity of new nodes when adding nodes to the original graph.
•	Since it has been shown that simply duplicating nodes with low degrees can improve the model performance [3], and the latent space method may have difficulty representing the real global graph structure [4], the improvement of the proposed method may come from the larger number of node features rather than the synthetic graph structure.

[1] Zhao T, Liu Y, Neves L, et al. Data augmentation for graph neural networks[C]//Proceedings of the aaai conference on artificial intelligence. 2021, 35(12): 11015-11023.
[2] Wang Y, Wang W, Liang Y, et al. Nodeaug: Semi-supervised node classification with data augmentation[C]//Proceedings of the 26th ACM SIGKDD International Conference on Knowledge Discovery & Data Mining. 2020: 207-217.
[3] Guo Z, Zhao T, Liu Y, et al. Node duplication improves cold-start link prediction[J]. arXiv preprint arXiv:2402.09711, 2024.
[4] Vignac C, Krawczuk I, Siraudin A, et al. DiGress: Discrete Denoising diffusion for graph generation[C]//Proceedings of the 11th International Conference on Learning Representations. 2023.

---

> ### Author Response · Authors · 2024-10-30
> **Response to Reviewer su2i Part 1**
>
> We appreciate the review and the suggestions in this review. The raised issues are addressed below.
>
> **1. “... labeling and imputing features and connectivity of new nodes when adding nodes to the original graph.”**
>
> **How DoG overcomes the challenges in labeling and imputing features.** To generate the labeled synthetic nodes, we employ the conditional Latent Diffusion Model (LDM) [1] based on the Classifier-Free Guidance (CFG) [2]. The class labels are converted to conditional embeddings, which are incorporated into the latent features of the synthetic nodes. In the generation process, we manually set the labels of the synthetic nodes to achieve controllable generation. Detailed descriptions of the conditional generation of the synthetic graph structures are presented at the beginning of Section 4.2 of our paper.
>
> **How DoG overcomes the challenges in the connectivity of new nodes.** To connect the synthetic nodes to the real nodes in the original graph, we propose a novel Graph Autoencoder (GAE), which encodes both the node attributes and the neighbor maps of the nodes into latent features. After generating the synthetic latent features with the LDM, the decoder of the GAE decodes the latent features into the node attributes and the neighbor maps of the synthetic nodes, which indicate the real nodes in the original graph that each synthetic node should connect to. Detailed descriptions of how the synthetic nodes and edges are incorporated into the original graph are presented at the end of Section 4.2 of our paper.
> Algorithm 2 in Section D of the appendix presents the details of the entire generation process of the augmented graph.

---

> ### Author Response · Authors · 2024-10-30
> **Response to Reviewer su2i Part 2**
>
> **2. “The authors could better justify whether the improvement in the proposed method comes from the increase in the number of nodes or the synthetic graph structures.”**
>
> To evaluate the effectiveness of the synthetic graph structures generated by the DoG in improving the performance of GNNs for node classification, we compare the DoG with two categories of ablation models of the DoG, which do not incorporate the synthetic graph structures generated by the DoG. The first category of the ablation models, which is referred to as DoG-Duplicate, only adds the synthetic nodes generated by DoG to the training set of the original graph without connecting them to the real nodes in the original graph. The second category of ablation models, which is referred to as DoG-KNN, connects the synthetic nodes generated by the DoG to the real nodes in the original graph using the K-Nearest Neighbor (KNN) algorithm on the node attributes. The value of $K$ in the KNN algorithm is set to {$1, \lceil d_{avg}/4 \rceil, \lceil d_{avg}/2 \rceil,  \lceil d_{avg} \rceil,  \lceil 2\times d_{avg} \rceil,  \lceil 4 \times d_{avg} \rceil$}, where $d_{avg}$ is the average node degree of the original real graph. The results are shown in the table below. EXPHORMER [3] is employed as the base model for semi-supervised node classification. It is observed that although increasing the number of nodes can boost the performance of the base GNN model, DoG with the connectivity of the new nodes, or the synthetic nodes, significantly outperforms the ablation model, DoG-duplicate, which does not have connectivity of the new nodes. For example, DoG outperforms DoG-Duplicate by 1.4\% in the node classification accuracy on the Citeseer dataset. Furthermore, DoG exhibits much better performance than DoG-KNN with different K values. For instance, the best-performing DoG-KNN model with $K=d_{avg}$ improves the classification accuracy of the base model from 72.9\% to 73.8\% on the Citeseer dataset, whereas the DoG model further elevates the classification accuracy to 74.8\%.
> In addition, we compare DoG with the NODEDUP [4], which enlarges the training set by duplicating real nodes with low degrees in the original graph. Following the settings in the paper [4], only the nodes with degrees not exceeding 2 are duplicated. Edges connecting the low-degree nodes and their own duplicates are created. It is observed that DoG also significantly outperforms the NODEDUP. For instance, DoG outperforms NODEDUP by 1.1\% in the node classification accuracy on the Citeseer dataset.
> The ablation study results underline the efficacy of the synthetic graph structures generated by DoG in boosting classification performance.
>
> | Methods |Cora |Citeseer | Pubmed| CoauthorCS| ogbn-arxiv|
> | :-- | :--: | :--: | :--: | :--: | :--: |
> | EXPHORMER |84.1 |72.9 |83.2 |95.7 |72.4 |
> | NODEDUP [4] | 84.7($\uparrow$0.6) | 73.7($\uparrow$0.8) | 83.6($\uparrow$0.4) | 95.8($\uparrow$0.1) | 72.7($\uparrow$0.3) |
> | DoG-Duplicate | 84.3($\uparrow$0.2) | 73.4($\uparrow$0.5) | 83.5($\uparrow$0.3) | 95.9($\uparrow$0.2) | 72.4($\uparrow$0.0) |
> | DoG-Duplicate | 84.3($\uparrow$0.2) | 73.4($\uparrow$0.5) | 83.5($\uparrow$0.3) | 95.9($\uparrow$0.2) | 72.4($\uparrow$0.0) |
> | DoG-KNN ($K=1$) | 84.8($\uparrow$0.7) | 73.7($\uparrow$0.8) | 83.6($\uparrow$0.4) | 96.2($\uparrow$0.5) | 72.8($\uparrow$0.4) |
> | DoG-KNN ($K=\lceil d_{avg}/4 \rceil$) | 84.8($\uparrow$0.7) | 73.6($\uparrow$0.7) | 83.8($\uparrow$0.6) | 96.2($\uparrow$0.5) | 72.7($\uparrow$0.3) |
> | DoG-KNN ($K=\lceil d_{avg}/2 \rceil$) | 84.7($\uparrow$0.6) | 73.7($\uparrow$0.8) | 83.9($\uparrow$0.7) | 96.3($\uparrow$0.6) | 72.8($\uparrow$0.4) |
> | DoG-KNN ($K=\lceil d_{avg} \rceil$) | 84.9($\uparrow$0.8) | 73.8($\uparrow$0.9) | 83.9($\uparrow$0.7) | 96.1($\uparrow$0.4) | 72.6($\uparrow$0.2) |
> | DoG-KNN ($K=\lceil 2\times d_{avg} \rceil$) | 84.5($\uparrow$0.4) | 73.6($\uparrow$0.7) | 84.0($\uparrow$0.8) | 96.4($\uparrow$0.7) | 72.6($\uparrow$0.2) |
> | DoG-KNN ($K=\lceil 4\times d_{avg} \rceil$) | 84.4($\uparrow$0.3) | 73.6($\uparrow$0.7) | 83.8($\uparrow$0.6) | 96.3($\uparrow$0.6) | 72.7($\uparrow$0.3) |
> | DoG (EXPHORMER) | **85.7**($\uparrow$1.6) | **74.8**($\uparrow$1.9) | **84.6**($\uparrow$1.4) | **96.9**($\uparrow$1.2) | **73.4**($\uparrow$1.0) |
>
> **References**
>
> [1] Rombach, Robin, et al. "High-resolution image synthesis with latent diffusion models." Proceedings of the IEEE/CVF conference on computer vision and pattern recognition. 2022.
>
> [2] Ho, Jonathan, and Tim Salimans. "Classifier-free diffusion guidance." arXiv preprint arXiv:2207.12598 (2022).
>
> [3] Shirzad, Hamed, et al. "Exphormer: Sparse transformers for graphs." International Conference on Machine Learning. PMLR, 2023.
>
> [4] Guo, Zhichun, et al. "Node duplication improves cold-start link prediction." arXiv preprint arXiv:2402.09711 (2024).

---

### Decision · Action_Editor_Pgrd · 2025-01-22

**Recommendation:** Accept with minor revision

**Comment:**

The paper was initially reviewed by three reviewers. The reviewers found the proposed method technically novel and showing good performance against baselines. However, they raised several important concerns on the initial version :
- Lack of recent related work, including recent GNNs for node classification and other discussions on graph data augmentation.
- Unclear motivation/justification for subgraph generation and adopting latent diffusion model design.
- Insufficient experimental settings and missing recent baselines in experiments.

The author rebuttal addressed most of these concerns satisfactorily by incorporating missing references, clarifying the motivation and technical design, and adding more ablation study and comparisons with recent baselines, including larger-scale graphs. As a result, two reviewers responded favorably on the revised draft (1 accept and 1 weak accept). Although the third reviewer did not decide on a final rating, the AE found the author's detailed response sufficiently addressed initial concerns, especially regarding missing recent baselines and lack of clarity.

Overall, while the empirical study in this work is limited to certain types of graph data, the efficacy of the proposed new augmentation strategy for node-level task learning is validated across a range of benchmarks. Therefore, the AE agrees with the majority of the reviewers and recommends acceptance for a revised version of this submission. The authors should incorporate the requested changes into the final version, including additional experimental comparisons and clarifications provided during the rebuttal.

**Audience:**

The paper addresses the challenge of data augmentation for node-level classification tasks in network data and develops a technique for subgraph generation, which will be of interest to researchers in generative modeling and network data analysis.

**Claims And Evidence:**

The paper presents a diffusion-based graph generation method that augments the original graph with synthetic subgraphs to improve training for node-level tasks. To generate the subgraphs, the authors develop a latent diffusion model of graph nodes and their connectivity with a Bi-level Neighborhood Decoder. Subsequently, the augmented graphs are used to train graph neural networks for node-level tasks, employing a low-rank regularization strategy to mitigate the impact of noise in the generated data.

The authors provided experimental evaluations on several public datasets and have made the code available. The results from the initial submission and rebuttal demonstrate consistent performance improvement over a range of baselines on semi-supervised node classification and graph contrastive learning tasks. The theoretical analysis on low-rank regularization and ablation study further underscore the effectiveness of the proposed augmentation strategy.

---

> ### Author Response · Authors · 2025-02-02
> **Thank you for your decision**
>
> Dear Action Editor,
>
> Thank you for your favorable decision on this paper, and we really appreciate your time handling and reviewing this paper. We will incorporate the changes/revisions along with the additional experimental results and clarifications made in the rebuttal phase into the final version.
>
> Best Regards,
>
> The Authors